# NELL2-Robo3 complex structure reveals mechanisms of receptor activation for axon guidance

Joseph S. Pak [1,2,6], Zachary J. DeLoughery [3,4,6], Jing Wang[1,2], Nischal Acharya[3,4], Yeonwoo Park[5], Alexander Jaworski [3,4,7 ✉] & Engin Özkan [1,2,7 ✉]

Axon pathfinding is critical for nervous system development, and it is orchestrated by molecular cues that activate receptors on the axonal growth cone. Robo family receptors bind Slit guidance cues to mediate axon repulsion. In mammals, the divergent family member Robo3 does not bind Slits, but instead signals axon repulsion from its own ligand, NELL2. Conversely, canonical Robos do not mediate NELL2 signaling. Here, we present the structures of NELL-Robo3 complexes, identifying a mode of ligand engagement for Robos that is orthogonal to Slit binding. We elucidate the structural basis for differential binding between NELL and Robo family members and show that NELL2 repulsive activity is a function of its Robo3 affinity and is enhanced by ligand trimerization. Our results reveal a mechanism of oligomerization-induced Robo activation for axon guidance and shed light on Robo family member ligand binding specificity, conformational variability, divergent modes of signaling, and evolution.

[1] Department of Biochemistry and Molecular Biology, University of Chicago, Chicago, IL 60637, USA. [2] Grossman Institute for Neuroscience, Quantitative Biology and Human Behavior, University of Chicago, Chicago, IL 60637, USA. [3] Department of Neuroscience, Division of Biology and Medicine, Brown University, Providence, RI 02912, USA. [4] Robert J. and Nancy D. Carney Institute for Brain Science, Providence, RI 02912, USA. [5] Department of Human Genetics, University of Chicago, Chicago, IL 60637, USA. [6] These authors contributed equally: Joseph S. Pak, Zachary J. DeLoughery. [7] These authors jointly supervised this work: Alexander Jaworski, Engin Özkan. ✉email: alexander_jaworski@brown.edu; eozkan@uchicago.edu

Establishing proper neuronal connectivity during embryonic development is critical for formation of a functional nervous system, and guidance of axons to their correct targets is a key step in the assembly of neural circuits. Axon pathfinding is orchestrated by attractive and repulsive molecular cues that activate receptors on the axonal leading process, the growth cone[1]. For example, Netrin-1, acting through its DCC family receptors, promotes axon growth and mediates attraction[2]; Slit proteins, on the other hand, are prototypical repulsive cues that signal through receptors of the Robo family[3,4]. Many classes of guidance cues and receptors, including Netrin/DCC and Slit/Robo, are conserved across taxa from invertebrates to mammals; however, evolutionary expansion of the relevant gene families and functional specialization of individual family members likely allow wiring of the more complex nervous systems found in higher organisms, including humans[5]. The full extent and nature of these modifications to the molecular toolkit for axon guidance remain elusive.

Robo family receptors are type I transmembrane proteins that instruct neuronal wiring in various organisms, including worms, flies, mice, and humans, in addition to fulfilling diverse functions outside the nervous system[3,4]. The N-terminal extracellular domains (ECDs) of neuronally expressed Robos are composed of five immunoglobulin-like (IG) and three fibronectin type III (FN) domains (Fig. 1a)[6], while the vertebrate-specific family member Robo4, which is not expressed in the nervous system and regulates angiogenesis, contains only two IG and two FN domains[7,8]. The first IG domain (IG1) of neuronal Robos binds to the canonical Robo ligands belonging to the Slit family[9,10]. Slits are multidomain secreted proteins (Fig. 1a)[11] that interact with

Robos through their second leucine-rich repeat (LRR) domain[10,12] to mediate axon repulsion, branching, and fasciculation[3]. How Robos are activated by Slits is a topic of contention. There are conflicting reports on whether Slit dimerization plays a role in Robo-mediated downstream effects[12–14], and if the oligomerization state of Robo changes upon Slit binding[15]. A recent study supported a mechanism involving Slit-induced conformational change of preformed Robo dimers to allow repulsive signaling[16], while another manuscript proposed two competing modes of Robo dimer formation, regulated via Slit binding[17]. A *trans* mode of Robo dimerization to turn off responses to Slit was previously proposed[18]. In addition, heparan sulfate binding by Slits is known to modulate Slit–Robo binding and signaling[19–21], but a complete molecular model for the activation of a Slit–Robo–Heparan sulfate complex remains to be established.

In mammals, there are three Slit paralogs (Slit1–Slit3) and three neuronally expressed Robos (Robo1–Robo3). While Robo1 and Robo2 function as classical Slit receptors, the divergent family member Robo3 does not bind Slits and instead acts as a negative regulator of Robo1/2-mediated Slit repulsion[22–25]. Robo3 also indirectly boosts axon attraction to Netrin-1 without interacting with Netrin-1 itself[26]. Further, a recently identified Robo3 ligand, NELL2, signals axon repulsion[27]. These three noncanonical functions of Robo3 are best understood in the guidance of commissural axons across the midline of the mouse spinal cord and hindbrain, which has long served as a prime model system for studying axon pathfinding mechanisms. The floor plate at the ventral midline expresses Netrin-1 and Slit1/2/3, while NELL2 is expressed in regions flanking the commissural axon trajectory, including the ventral horn[27–29]. In commissural

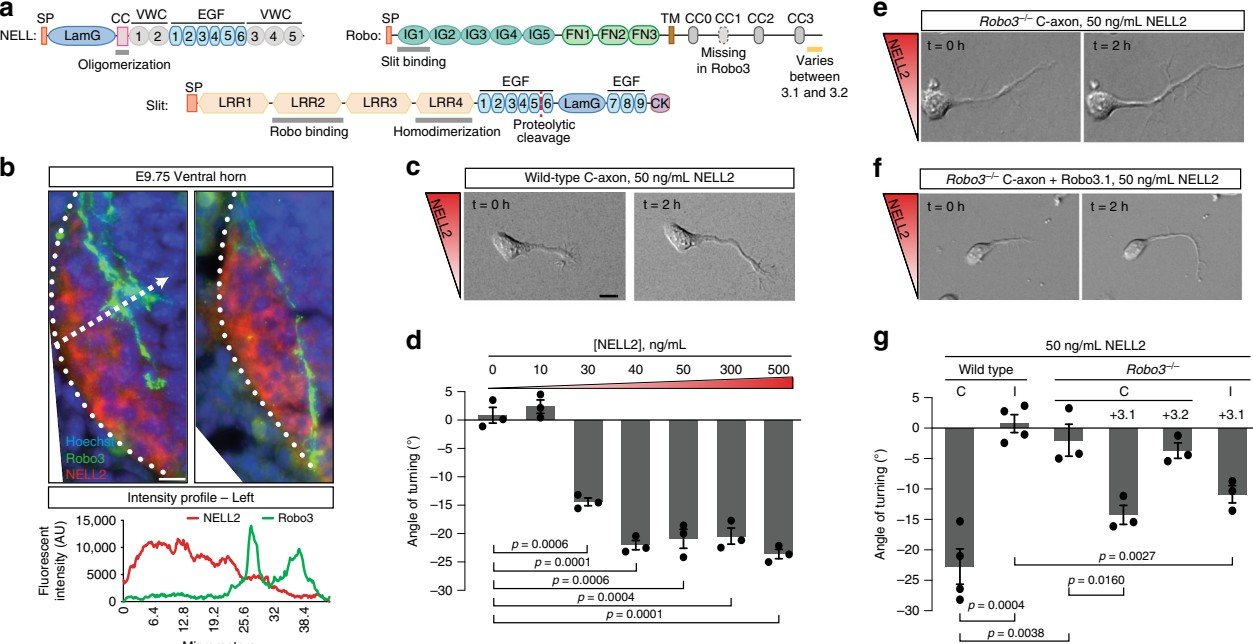

**Fig. 1 Robo3.1 is necessary and sufficient for axon repulsion from NELL2. a** Domain structure of NELLs, Robos, and Slits, drawn true to scale for human NELL2, Robo3, and Slit1. CC with numbers indicate conserved cytoplasmic motifs. SP signal peptide. TM transmembrane helix. CK C-terminal cysteine knot. **b** Transverse spinal cord sections from E9.75 mouse embryos were labeled with Hoechst stain and antibodies against Robo3 and NELL2. Robo3-expressing precrossing commissural axons grow around areas of NELL2 expression in the ventral horn. Dotted line indicates spinal cord border. Fluorescence intensity profile (bottom) was generated by line scan along the dashed arrow. **c** DIC images of a commissural axon turning away from a gradient of NELL2 in a Dunn chamber (0 and 2 h). **d** Quantification of commissural axon turning angles for increasing concentrations of NELL2 (*n* = 3 independent experiments for all conditions). **e, f** DIC images of *Robo3*−/− neurons in a NELL2 gradient. *Robo3*−/− axons do not turn in response to NELL2 (**e**), unless rescued with Robo3.1 (**f**). **g** Quantification of turning angles in response to NELL2 for commissural (labeled C) axons (wild type or *Robo3*−/− with or without Robo3.1 or Robo3.2 rescue) or ipsilateral (labeled I) axons (*n* = 4 for wild-type commissural and ipsilateral axons, *n* = 3 for all other conditions). Scale bar, 10 µm (**b**, **c**, **e**, **f**). Error bars indicate SEM.

neurons, Robo3 exists in two splice isoforms that differ at the C-termini of their intracellular domains (ICDs): Robo3.1 is expressed before midline crossing, and Robo3.2 is expressed on postcrossing axons[30]. Robo3.1 appears to facilitate midline crossing by preventing premature Slit repulsion from the floor plate, promoting Netrin-1-induced growth to the midline, and mediating NELL2 repulsion from the ventral horn, while Robo3.2 promotes midline exit after crossing through unknown mechanisms[22,26,27,30]. Consequently, in mice lacking *Robo3*, all commissural axons fail to reach the floor plate and instead project through the ipsilateral ventral horn[22,27]. This crucial role of Robo3 as a signaling hub in midline guidance is conserved in humans, as midline crossing defects are seen in patients with *ROBO3* mutations, who exhibit Horizontal Gaze Palsy with Progressive Scoliosis[31]. Even though Robo3 is required to mediate NELL2 repulsion in commissural axons and both Robo3.1 and Robo3.2 bind NELL2, only precrossing, not postcrossing, axons are repelled by NELL2[27]. It has remained unclear whether this differential NELL2 responsiveness reflects distinct signaling capabilities of Robo3.1 and Robo3.2 or other differences between pre- and postcrossing axons.

NELL2 is a secreted glycoprotein containing a laminin G-like (LamG) domain, a coiled coil (CC) domain, and six epidermal growth factor-like (EGF) domains, flanked by five von Will-ebrand factor type C (VWC) domains (Fig. 1a)[32–34]. The interaction between NELL2 and Robo3 maps to the EGF domains in NELL2 and the FN domains in Robo3[27]. In addition to commissural axon guidance, NELL2 controls retinal ganglion cell wiring[35,36], and it is expressed in various regions of the developing nervous system[33,37,38]. The additional mammalian NELL family member, NELL1, also binds Robo3, but it is not expressed in the spinal cord and has little or no repulsive activity for commissural axons[27]. Instead, NELL1 has been implicated in osteogenesis through as yet unidentified mechanisms[39]. While NELLs bind Robo3, they do not strongly interact with full-length Robo1/2[27]. The exact nature of the NELL1/2–Robo3 complex interface, the basis of the different Robo1/2 and Robo3 ligand binding specificities, and the mechanism of NELL2-dependent Robo3 activation for axon repulsion are unknown.

Here, we investigate NELL–Robo3 complex signaling in axon guidance. We identify Robo3.1 as the only neuronal Robo that can function as a NELL receptor for commissural axon repulsion. Further, we show that NELL1/2–Robo3 binding is mediated by the combined interaction of NELL EGF2 and EGF3 domains with the Robo3 FN1 domain, and we report the crystal structures of NELL1–Robo3 and NELL2–Robo3 complexes. We demonstrate that reducing NELL2 affinity for Robo3 decreases its axon guidance activity, while NELL1 possesses axon repulsive activity orders of magnitude lower than NELL2 despite comparable Robo3 affinities. We also find that divergence of the FN1 domain between Robo family members underlies their different NELL affinities and signaling capabilities, while conformational variability in the Robo ECDs further regulates accessibility of the NELL-binding site. These findings are consistent with recently published manuscripts showing a cryptic NELL-binding site on Robo2[40], possibly occluded by a hairpin-like conformation observed in the Robo2 ECD[17]. Lastly, we show that NELL-mediated oligo-merization of Robo3 monomers is a strong contributor to NELL–Robo3 signaling and axon repulsion. Together, these findings expand on models of Robo activation by guidance cues and indicate that mammalian Robo3 underwent evolutionary specialization in terms of its ligand binding specificity, con-formational landscape, and oligomerization state for activation and signaling.

## Results

**The Robo3.1 splice variant mediates NELL2 axon repulsion.** Commissural axons express the Robo3 splice variant Robo3.1 exclusively before midline crossing and replace this isoform with Robo3.2 after crossing[30]. Even though both isoforms can bind NELL2, only precrossing commissural axons are repelled by NELL2, but postcrossing axons are not, thereby mirroring the mutually exclusive presence of Robo3.1 and Robo3.2[27]. We revisited the roles of the two Robo3 splice isoforms in NELL2 signaling. We first examined the relationship of pre-crossing axons and the NELL2 protein expression domain in mouse embryonic day 9.75 (E9.75) spinal cord sections by immunohistochemistry and found that the first Robo3-expressing axons extend toward the midline by precisely circumnavigating a sharply delineated NELL2-positive region in the ventral horn (Fig. 1b). This result indicates that NELL2 forms a very steep gradient in the neural tube and supports the idea that Robo3.1-mediated NELL2 repulsion steers precrossing pioneer axons along the border formed by this gradient.

To directly test the roles of Robo3.1 and Robo3.2 in axon guidance by NELL2, we examined these isoforms for their ability to mediate NELL2 repulsion in vitro. We adapted Dunn chamber axon turning assays[41] to study responses of E11.5 mouse dorsal spinal cord neurons to gradients of NELL2 (Supplementary Fig. 1a) and found that NELL2 induces robust repulsion of Robo3.1-positive precrossing commissural axons that plateaus at a peak concentration of 40 ng/ml (Fig. 1c, d, Supplementary Fig. 1b–d, and Supplementary Movie 1). Ipsilaterally projecting neurons, which do not express Robo3[22], and commissural neurons from *Robo3*[−/−] embryos do not respond to NELL2 (Fig. 1e, g, Supplementary Fig. 1e, and Supplementary Movie 2). When we reintroduced either Robo3.1 or Robo3.2 into *Robo3*[−/−] commissural neurons via cDNA electroporation (Supplementary Fig. 1a, f), we found that only Robo3.1 can restore NELL2-dependent axon repulsion (Fig. 1f, g and Supplementary Fig. 1g), even though both isoforms are expressed on the growth cone surface after electroporation (Supplementary Fig. 1h). Further, forced Robo3.1 expression in ipsilaterally projecting neurons also induces NELL2-dependent axon repulsion in these normally NELL2-unresponsive cells (Fig. 1g). Thus, Robo3.1 is both necessary and sufficient for NELL2-induced axon repulsion in dorsal spinal cord neurons, while Robo3.2 cannot mediate NELL2 signaling. The Robo3.1 isoform was therefore selected for additional functional studies.

**NELL2 EGF domains 2 and 3 bind the Robo3 FN1 domain.** We have previously shown that the EGF domains of NELL2 and the FN domains of Robo3 are sufficient for binding[27]. To understand how the NELL2–Robo3 complex forms, we further mapped the domains necessary for the ligand–receptor interaction. Binding of human (h) NELL2-alkaline phosphatase (AP) fusions to hRobo3.1-expressing COS-7 cells revealed that NELL2 EGF2 and EGF3 together, but not in isolation, bind Robo3 FN1, but not any of the other Robo3 FN domains (Fig. 2a, b). NELL2–Robo3 binding is therefore mediated by NELL2 EGF2/3 and Robo3 FN1.

We next set out to determine the structure of NELL2 in complex with Robo3. Using constructs that include the hNELL2 EGF domains 1–6 and the hRobo3 FN domains 1–3, we obtained a stable complex of these molecules, as demonstrated by size-exclusion chromatography (SEC) (Fig. 2c). Surface plasmon resonance (SPR) analysis of this complex yielded a dissociation constant of $0.6\,\mu M$ (Fig. 2d, e). We crystallized a complex of NELL2 EGF1–6 with Robo3 FN1, and Robo3 FN2–3 alone, following proteolysis within the 15-residue-long flexible linker between FN1 and FN2 in crystallization drops (Supplementary

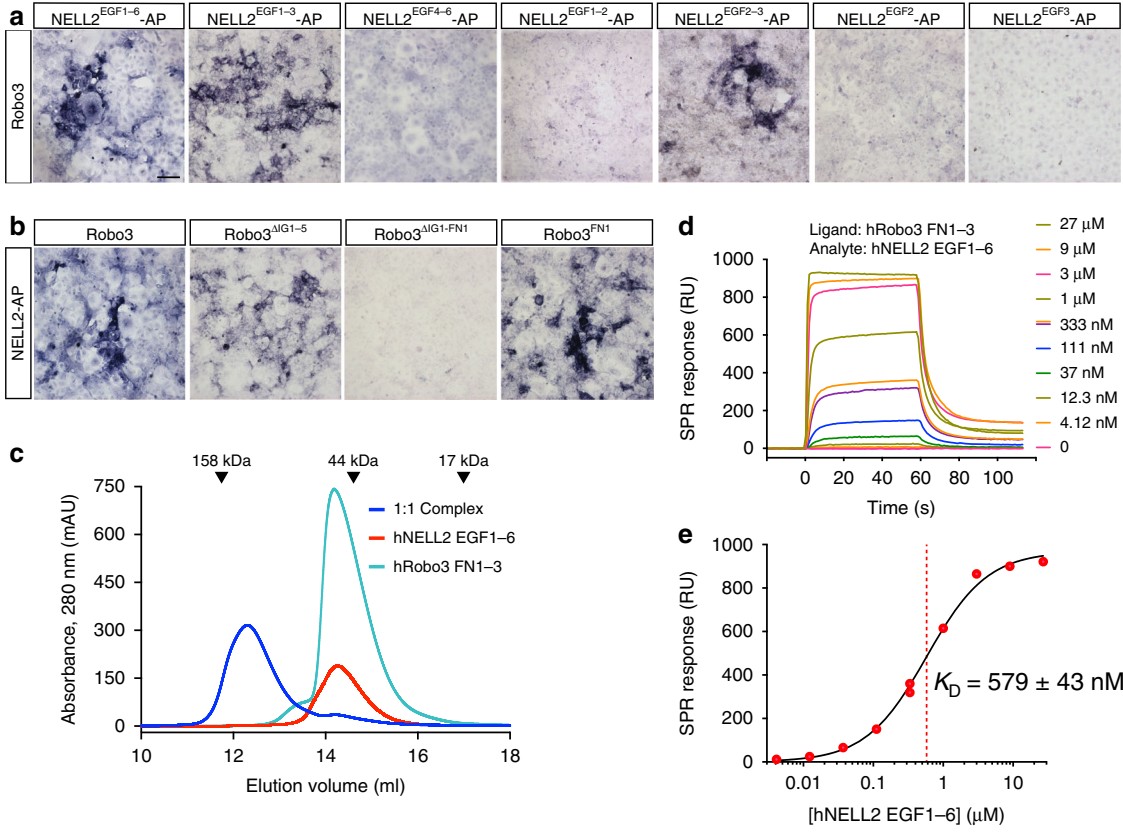

**Fig. 2 NELL2 EGF2–3 bind Robo3 FN1 to mediate complex formation. a, b** Domains mediating NELL2–Robo3 interactions were mapped using a COS-7-based AP-fusion protein binding assay. NELL2 EGF2 and EGF3 together are necessary and sufficient for binding to Robo3 (**a**). Robo3 FN1 is necessary and sufficient for NELL2 binding (**b**). **c** Size-exclusion chromatography of the hNELL2 EGF1–6 and hRobo3 FN1–3 complex. hNELL2 EGF1–6, hRobo3 FN1–3, and a molar 1:1 mixed complex samples were injected in a Superdex 200 Increase 10/300 column, and the elution profile was recorded by following absorbance at 280 nm with a pathlength of 0.2 cm. Dark blue: hNELL2 EGF1–6 + hRobo3 FN1–3 complex sample; Cyan: hRobo3 FN1–3; and Red: hNELL2 EGF1–6. AU: Absorbance units. SPR sensorgrams (**d**) and equilibrium response fitting to a Langmuir 1:1 binding model (**e**) for biotinylated hRobo3 FN1–3 as ligand on an NLC/neutravidin chip with hNELL2 EGF1–6 as the analyte (mobile phase) collected on a ProteOn XPR36. Legend on the right in (**d**) refers to concentration of the analyte injected on the SPR chip. Black curve in (**e**) is the Langmuir model fit to response values measured at equilibrium. ± refers to standard error of the fit. Scale bar, 100 μm, (**a**, **b**).

Fig. 2a). The NELL2–Robo3 complex structure was determined by an MR-SAD phasing strategy to 2.76-Å resolution (Supplementary Table 1). The structure shows direct binding between the NELL2 EGF2 and EGF3 domains and the Robo3 FN1 domain (Fig. 3a), consistent with our domain mapping data (Fig. 2a, b). The two EGF domains wrap around the FN domain and interact with the β-sheet composed of the *CDFG* strands (Fig. 3a). To our knowledge, this mode of interaction between EGF and FN domains has not been observed before. Netrin EGF domains and DCC FN domains adopt entirely different geometries in their complexes (Supplementary Fig. 2b)[42,43], and the NELL2–Robo3 complex therefore exhibits no structural similarity to and shares no common ancestry with the Netrin–DCC complex. We also determined the crystal structure of hRobo3 FN2–3, which showed two linearly extended FN domains with no unexpected features for the FN fold (Supplementary Fig. 2c).

In the crystal structure of the NELL2–Robo3 complex, we observed electron density for three calcium ions bound to the EGF2, EGF5, and EGF6 domains (Supplementary Fig. 2d–f). The calcium coordination sites are conserved among vertebrate and invertebrate NELL sequences (Supplementary Fig. 2d–f). Since we did not include calcium in our purification and crystallization buffers, these calcium ions must be held tightly by the EGF domains. The electron density also showed one O-linked glycan residue on the NELL2 EGF4 domain in a motif known to be

recognized by a dedicated transferase in several signaling and matrix EGF domain proteins, including Notch and NELL1[44,45], raising the possibility of shared origins or adopted functions for NELLs and Notch (Supplementary Fig. 2g).

**NELL2–Robo3 affinity dictates axon repulsion from NELL2.** The human NELL2–Robo3 complex structure displays a surprising partitioning of the overall binding interface into a highly polar interface at the EGF2–FN1 contact (Fig. 3b), and an entirely hydrophobic interface at the EGF3–FN1 contact (Fig. 3c). To confirm the crystallographically observed complex structure, we engineered single- and double-point interface mutants of hNELL2 EGF1–6 and hRobo3 FN1–3, and tested them for binding in flow cytometry-based cell staining assays. Several mutations nearly abolish binding, while many others yield diminished binding activities, confirming the validity of the crystallographic model (Fig. 3d, e and Supplementary Fig. 3a, b). Cell staining results were further corroborated using the high-throughput, ELISA-like extracellular interactome assay (ECIA)[46] (Supplementary Fig. 3c). To be able to test the functional effects of these mutations in an experimental setting using mouse commissural neurons, we also measured affinities of NELL2 interface mutants by SPR analysis of mouse (m) NELL2 EGF1–6 and Robo3 FN1–3 (Fig. 4a, b and Supplementary Fig. 3d). The binding results for mouse and

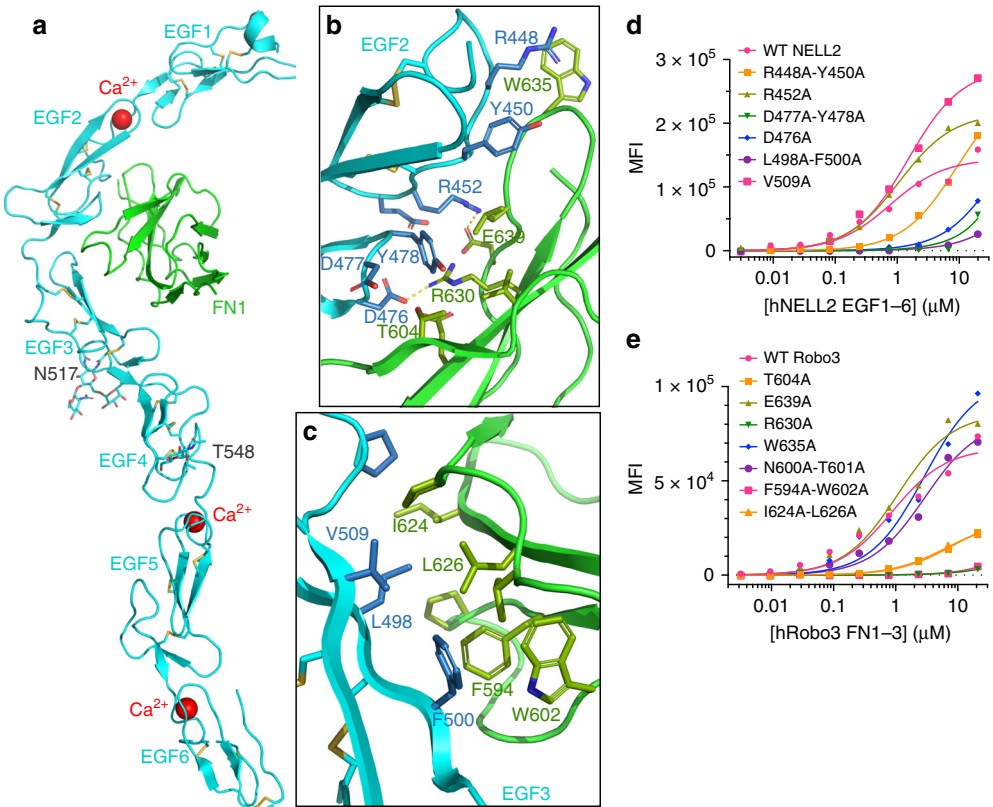

**Fig. 3 Crystal structure of the NELL2–Robo3 complex. a** The structure of hRobo3 (green) bound to hNELL2 (cyan). The three calcium ions are depicted as balls; the two glycan moieties and the side chains they are linked to are shown as sticks. **b** Interactions of the EGF2 domain with Robo3 are dominated by polar contacts. **c** Interactions of the EGF3 domain with Robo3 are exclusively hydrophobic. **d** Binding isotherms for biotinylated wild type and mutant hNELL2 EGF1–6 measured against S2 cells expressing hRobo3 FN1–3. *y*-axis represents mean fluorescence intensity (MFI) from fluorescent streptavidin. **e** Binding isotherms for biotinylated wild type and mutant hRobo3 FN1–3 measured against S2 cells expressing hNELL2 EGF1–6. Apparent $K_D$ values for S2 cell staining experiments are listed in Supplementary Fig. 3a.

human NELL2 and Robo3 using three different methods are in full agreement and confirm the structural model of the complex in solution.

To assess the necessity of the NELL2–Robo3 interface for NELL2-mediated axon repulsion, we tested the guidance activity of mNELL2 point mutants using the Dunn chamber axon guidance assay (Fig. 4c–e). We found that mutants with mildly (2.4-fold) reduced mRobo3 affinity (V509A and R452A) repel commissural axons as strongly as wild-type mNELL2 (Fig. 4c, e). The R448A/Y450A double mutant, which weakened affinity by 120-fold, exhibits significantly reduced repulsive activity, while the L498A/F500A double mutant with 500-fold loss of binding fails to effect any detectable axon repulsion (Fig. 4d, e). Thus, the affinity of the NELL2–Robo3 interaction determines NELL2 axon guidance activity.

**NELL1 binds Robo3 and has weak axon repulsive activity.** NELL1 can bind Robo3, but it does not mediate strong commissural axon repulsion in vitro[27]. How NELL1 affinity for Robo3 relates to its guidance activity, has remained elusive. Using SEC, we isolated a hNELL1 EGF1–3–hRobo3 FN1 complex (Supplementary Fig. 4a). We crystallized and determined the structure of this complex at 1.8-Å resolution using molecular replacement (MR) with the NELL2–Robo3 model (Supplementary Table 1). The NELL1–Robo3 structure (Supplementary Fig. 4b) is highly similar to the NELL2–Robo3 complex, indicated by a root-mean square deviation of only 1.39 Å over all 225 Cα atoms in the EGF1–3 and FN1 domains (Fig. 5a). Using SPR, we found that

hNELL1 and hNELL2 EGF1–3 domains have comparable affinities for the hRobo3 FN1–3 domains (Supplementary Fig. 4c–e). Further, the interacting residues identified at the Robo3 interfaces for NELL1 and NELL2 are conserved among chordate NELL sequences, but not in protostome NELLs (Fig. 5b), which raises the possibility that invertebrate NELLs may not interact with Robos.

NELL1 EGF2 also has a calcium ion bound (Fig. 5a and Supplementary Fig. 4f) at the conserved site observed for NELL2 (Supplementary Fig. 2d). The calcium ion is not directly located at the Robo3-binding site, but is still within 11 Å of the closest Robo3 atom. Therefore, we examined NELL EGF1–3 binding to Robo3 FN1–3 with calcium chelation by EDTA, which only caused a modest (threefold to sixfold) reduction of Robo3 affinity for both NELL1 and NELL2 (Supplementary Fig. 4g–i). Taken together, these results indicate that NELL1 and NELL2 display no significant biochemical differences in Robo3 recognition.

Given the similarities in NELL1 and NELL2 domain structures and Robo3 recognition, we revisited the ability of NELL1 to repel axons. We investigated commissural axon responses to NELL1 in vitro and found that, at a peak concentration of 50 ng/ml, which produces maximum axon repulsion from NELL2 (Fig. 1d), NELL1 does not cause a turning response (Fig. 5c and Supplementary Fig. 4j). Weak repulsion was observed with 250 ng/ml NELL1, and maximum axon turning occurs between 1 and 2 µg/ml (Fig. 5c and Supplementary Fig. 4k), but the magnitude of this response never reaches the level seen with NELL2 (compare Figs. 1d and 5c). Overall, our results indicate that NELL1 can signal through Robo3 as its receptor to induce commissural axon

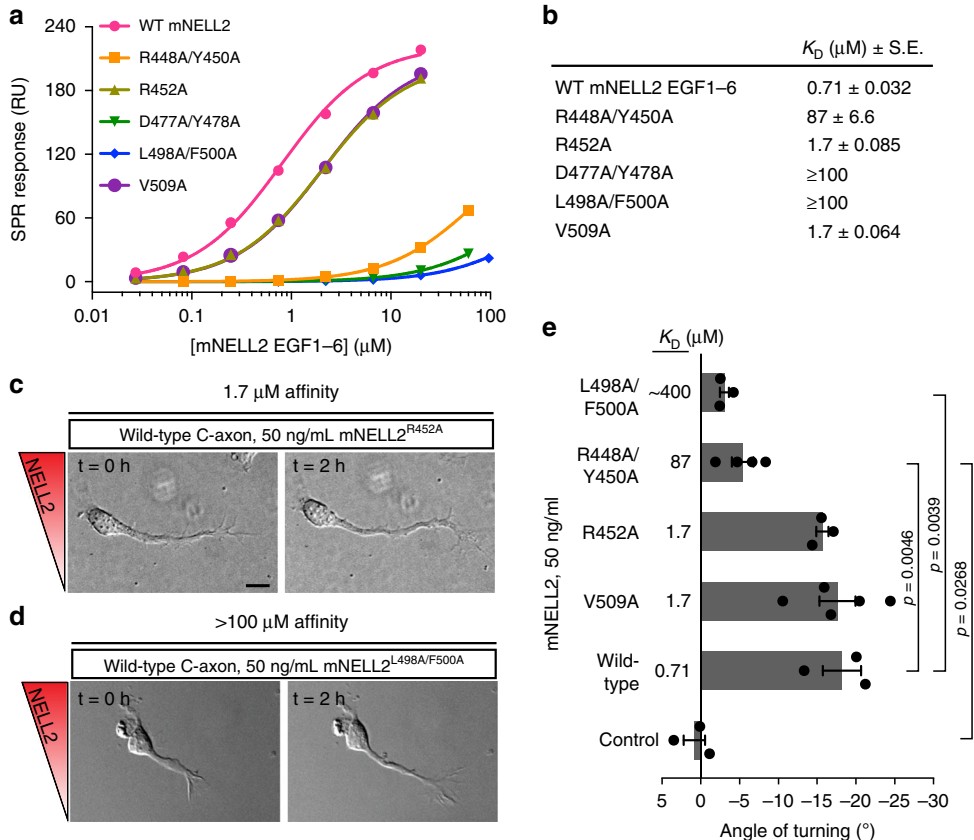

**Fig. 4 Axon repulsive activity of NELL2 correlates with its affinity towards Robo3. a, b** Binding isotherms (**a**) for SPR experiments testing the interaction of mRobo3 FN1–3 with mNELL2 EGF1–6 WT and mutants. Original sensorgrams are in Supplementary Fig. 3d. SPR responses are fit with 1:1 Langmuir isotherm model to calculate dissociation constants ($K_D$) (**b**). The ± errors represent standard error of the fit. DIC images of commissural axons exposed to mutant forms of NELL2 with Robo3-binding affinities of 1.7 μM (**c**) and ≥100 μM (**d**) (0 and 2 h). Commissural axons respond to NELL2 mutants with 1.7 μM binding affinity, but not with ≥100 μM affinity. (**e**) Quantification of commissural axon turning angles in response to wild-type NELL2 or NELL2 containing point mutations ($n = 3$ for WT NELL2, R452A, and L498A/F500A; $n = 4$ for R448A/Y450A; $n = 5$ for V509A). $K_D$ values are dissociation constants measured for mNELL2 mutants via SPR (**b**). Scale bar, 10 μm (**c**, **d**).

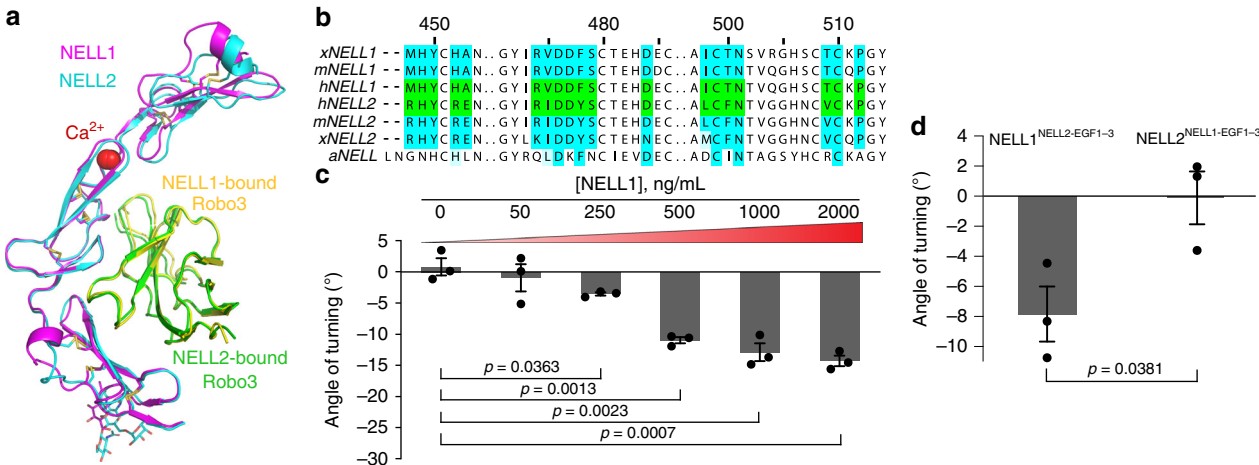

**Fig. 5 Axon repulsive activities and interactions of NELL family members with Robo3 are conserved to varying degrees. a** Crystal structure of hRobo3 FN1 bound to hNELL1 EGF1–3 (purple and yellow), overlaid with the hRobo3-hNELL2 structure (cyan and green). **b** Partial alignment of NELL sequences from *Xenopus laevis* (*x*), mouse (*m*), human (*h*), and the arthropod *Aedes aegypti* (*a*). Green squares indicate NELL residues at the Robo3-binding interface, which are highly conserved, except in aNELL. Blue positions are identical to human NELL1 or NELL2, while light blue represents conservative substitutions. Sequence numbering above the alignment is for hNELL2. **c** Quantification of commissural axon turning angles for increasing concentrations of NELL1 ($n = 3$ for all conditions). **d** Quantification of commissural axon turning angles in response to 50 ng/ml NELL1/2 chimeric molecules ($n = 3$ for all conditions). Error bars represent SEM.

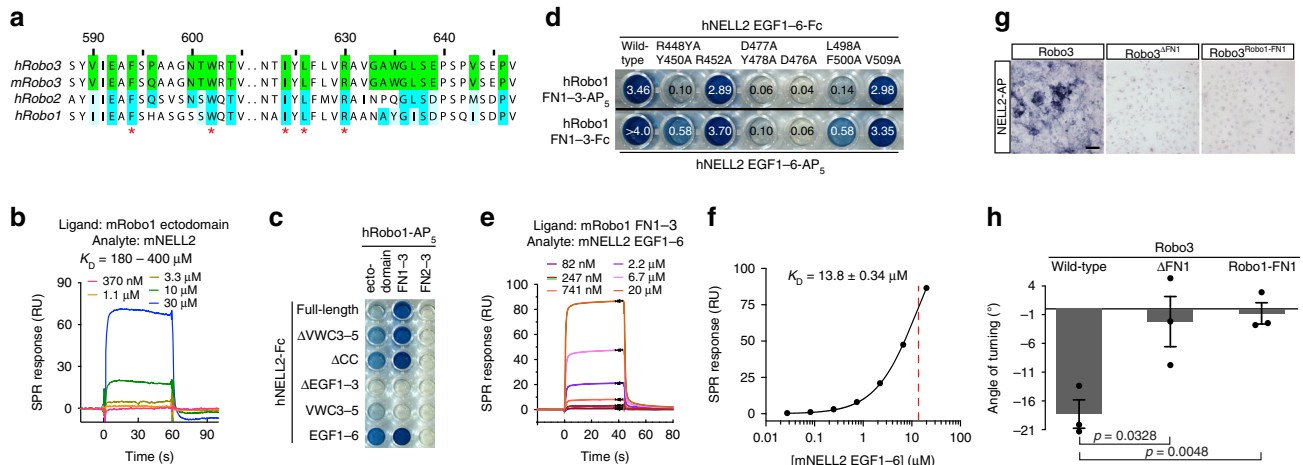

**Fig. 6 Robo1 interacts weakly with NELL2. a** Partial alignment of Robo FN1 sequences. Green squares indicate Robo3 residues at the NELL2-binding interface. Blue positions are identical in Robo1 and Robo2, while light blue represents conservative substitutions. **b** Surface plasmon resonance sensorgrams for the mRobo1 ectodomain-mNELL2 interaction. Expected maximal response is ~1,100 RU. **c** ECIA results for hRobo1 and hNELL2 show that FN1 and the EGF1–3 are necessary for the interaction. See Supplementary Fig. 6a for expression levels of each construct used. **d** ECIA results for Robo1 binding to NELL2 mutants designed at the Robo3-interaction interface. Comparison with Robo3 binding to NELL2 mutants in Supplementary Fig. 3c shows that Robo1 binding has the same binding surface on NELL2 with similar energetics of binding. Expression levels of ECIA constructs are measured with western blotting in Supplementary Fig. 6b. **e** Surface plasmon resonance sensorgrams for the mRobo1 FN1–3-mNELL2 EGF1–6 interaction. **f** Langmuir binding isotherm for (**e**). Dashed red line indicate the dissociation constant, $K_D$. Calculated $R_{max}$ is 143 R.U. The fit to the isotherm is reliable given a theoretical $R_{max}$ of 218 R.U. assuming a 100% active chip surface. **g** NELL2-AP-binding assay with cells expressing Robo3.1, Robo3$^{\Delta FN1}$, or Robo3$^{Robo1-FN1}$. **h** Quantification of turning angles in response to NELL2 for $Robo3^{-/-}$ commissural axons rescued with wild-type Robo3.1, Robo3$^{\Delta FN1}$, or Robo3$^{Robo1-FN1}$ ($n = 3$ for all conditions). Scale bar, 100 μm (**g**). Error bars indicate SEM.

repulsion, albeit less effectively than NELL2. To understand why NELL1 and NELL2 have different guidance activities despite the comparable affinities of their EGF1–3 domains for Robo3, we created chimeras in which the EGF1–3 domains were swapped between mNELL1 and mNELL2. Both chimeric proteins can interact with the Robo3 FN1–3 domains (Supplementary Fig. 5a, b). We examined the axon guidance activity of these chimeric NELLs and found that NELL1 containing NELL2 EGF1–3 (NELL1$^{NELL2-EGF1-3}$) repels commissural axons at 50 ng/ml, although not as strongly as NELL2, while NELL2 containing NELL1 EGF1–3 (NELL2$^{NELL1-EGF1-3}$) is not a potent axon repulsive cue at this concentration (Fig. 5d and Supplementary Fig. 5c, d). These results implicate the EGF1–3 domains as the major determining elements for differential axon repulsion activity of NELL1 and NELL2, despite the fact that we observed no significant differences in Robo3 affinities for the NELL1 and NELL2 EGF domains.

**Robo1 weakly interacts with NELL2.** While Robo3.1 is necessary and sufficient for NELL2-mediated axon repulsion (Fig. 1e–g), Robo1 and Robo2 are not required for NELL2 signaling in commissural axons[27]. If Robo–NELL interactions depend on the FN1 domain of Robos, we expect that the sequence of FN1 determines which Robos can function as NELL receptors. Surprisingly, the energetic hot spots for NELL2 binding on the hRobo3 interface, R630, F594/W602, and I624/L626 (Fig. 3d, e, and Supplementary Fig. 3a, b), are conserved in both the Robo1 and the Robo2 FN1 domains (Fig. 6a), suggesting that Robo1 and Robo2 may interact with NELLs. We were indeed able to detect weak binding activity using highly sensitive methods, SPR and ECIA, between the hRobo1 ectodomain and full-length hNELL2 with a $K_D \approx 350$ μM (Fig. 6b, c and Supplementary Fig. 6a). Thus, Robo1 binds NELL2 with about three orders of magnitude lower affinity than Robo3. As expected, this weak interaction is mediated by the same domains and amino acids as Robo3–NELL2 binding (Fig. 6c, d and Supplementary Fig. 6b). Interestingly, in

isolation, Robo1 FN1–3 exhibits stronger binding to mNELL2 EGF1–6 ($K_D = 14$ μM) (Fig. 6e, f), suggesting steric occlusion of the binding interface in full-length complexes. It should be noted, however, that even this increased, unmasked affinity between NELL2 and Robo1 is more than one order of magnitude weaker than the NELL2–Robo3 affinity, and binding might therefore not be sufficiently strong to effect NELL2 signaling in a physiological context. Overall, our results support a model of open vs. closed conformational states in full-length Robo1 (see Supplementary Fig. 6c for a schematic of this model). Furthermore, the ability of NELLs to bind to multiple mammalian Robos supports the idea that an ancestral Robo in the deuterostome lineage may have been a high-affinity receptor for both Slits and NELLs.

To further test the centrality of the Robo FN1 domain for recognizing NELL2 as a ligand, we created a mutant form of mRobo3.1 lacking FN1 (Robo3$^{\Delta FN1}$) and a chimeric version containing mRobo1's FN1 in place of its own (Robo3$^{Robo1-FN1}$). Using NELL2-AP binding assays, we found that both of these molecules fail to strongly interact with NELL2 (Fig. 6g), confirming the necessity of the Robo3 FN1 domain for high-affinity ligand binding. When we tested the activity of these mutant forms of Robo3 in NELL2-mediated axon guidance, we found that neither variant can rescue NELL2-induced axon turning of $Robo3^{-/-}$ commissural neurons in vitro (Fig. 6h and Supplementary Fig. 6d–f), further establishing importance of the Robo3 FN1 domain for NELL2 repulsive signaling. These results support the idea that evolutionary divergence of Robo3's and Robo1/2's FN1 domains was a key step in Robo3's specialization as a functional NELL2 receptor.

**Robo3 ECD is monomeric and assumes an open conformation.** Signaling by many type I transmembrane proteins, including Robo1, depends on receptor and ligand oligomeric states. Recent reports suggest that the Slit-binding paralogs, Robo1 and Robo2, may be dimers or compact tetramers[16,47]. Using analytical ultracentrifugation (AUC) and SEC, we confirmed that the Robo1

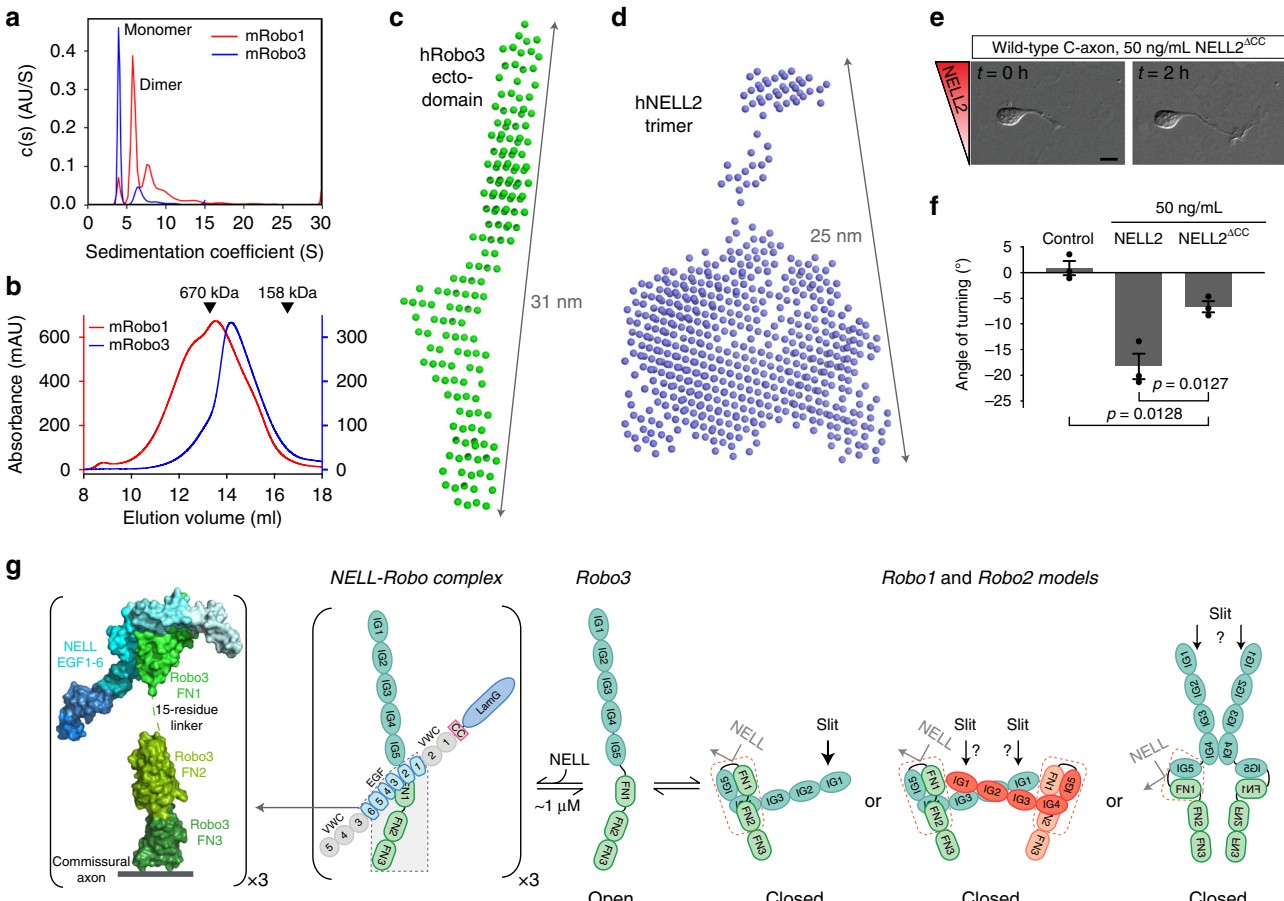

**Fig. 7 NELL2 trimers signal through multimerization of Robo3 monomers.** Sedimentation velocity (SV) AUC (**a**) and SEC (**b**) show that mRobo1 ectodomain is mostly dimeric (red curves), while mRobo3 ectodomain is a monomer (blue curves). Single-concentration SV runs (**a**) were performed with 2.5 μM Robo1 and 1.8 μM Robo3. See Supplementary Fig. 7 for a series of concentrations, which shows that mRobo1 ectodomain is in an equilibrium of monomer to dimer to oligomer, unlike the stably monomeric mRobo3. **c** Bead models calculated from SAXS data indicate a fully extended shape for hRobo3 ECD. Guinier plot for SAXS data is in Supplementary Fig. 8a. Pair distance distribution (*P*(*r*)) analysis is in agreement with the extended shape of Robo3. **d** Bead models calculated from SAXS data for hNELL2 indicate a trimer more compact than Robo3 ECD (**c**). Guinier plot analysis and *P*(*r*) plot are in Supplementary Fig. 9a. **e** DIC images of commissural axons exposed to wild-type NELL2 or NELL2[ΔCC] (0 and 2 h). **f** Quantification of axon turning angles in response to NELL2 and NELL2[ΔCC] (*n* = 3 for all conditions). NELL2[ΔCC] repels axons to a lesser degree than wild-type NELL2. **g** A model for conformational changes in mammalian Robos. Open Robo3 prefers to bind trimeric NELLs, while multiple conformational models exist for Robo1 and Robo2[16,17], which are in "closed" states for NELL binding. The conformational flexibility is created by the IG4-IG5, IG5-FN1, and FN1-FN2 linkers. The lightly shaded area in the NELL–Robo complex model was structurally characterized in this study (shown on the left side of the model). "x3" labels indicate trimerization via the CC domain of NELLs. Scale bar, 10 μm (**e**).

ectodomain exists primarily as a dimer, but in equilibrium with monomer and higher-order oligomers, while Robo3 ECD stays mostly monomeric at similar concentrations (Fig. 7a, b and Supplementary Fig. 7a–d). The predominantly monomeric nature of Robo3 was further upheld using multi-angle light scattering (MALS) experiments (Supplementary Fig. 7e), and small-angle X-ray scattering (SAXS) experiments show fully extended Robo3 ectodomain monomers (Fig. 7c and Supplementary Fig. 8a).

Overall, our results show that Robo1 (1) is an oligomer when unliganded, and (2) prefers a conformation where the NELL-binding site is occluded (a closed state). We designed a Robo construct where the FN1 domain from the predominantly open, monomeric mRobo3 was installed in mRobo1. This chimeric construct, mRobo1[Robo3-FN1], behaved as a pure monomer with an elution volume larger than both mRobo1 and mRobo3 in SEC experiments (Supplementary Fig. 8b). This smaller apparent size is not due to compaction of mRobo1[Robo3-FN1], as it still appears as a highly elongated molecule in SAXS experiments, as also observed for Robo1 and Robo3 (Supplementary Fig. 8c–f). When

we measured binding of mNELL2 to mRobo1[Robo3-FN1], we observed NELL2 affinity comparable with, if not slightly stronger than that of mRobo3 (Supplementary Fig. 8g–j). These biophysical observations indicate that the FN1 domain can dictate both the oligomeric state and open vs. closed conformation of Robos with regards to the NELL-binding interface.

**NELL2 trimerization promotes its repulsive activity.** Slits may exist as dimers[12,13]. For a comparison of the two Robo ligands, we investigated the oligomeric state of NELLs. NELLs had been reported to trimerize via disulfide linkages within or near their CC domain, as detected using nonreducing, denaturing polyacrylamide gels[32,48], but the oligomeric states of folded NELLs in solution had not been characterized before. Using a combination of SAXS and MALS experiments, we determined that both NELL1 and NELL2 are trimers in solution (Fig. 7d and Supplementary Fig. 9a, b). As expected, deletion of the CC region with its four cysteines (NELL2[ΔCC]) monomerizes NELL2 on

nonreducing denaturing gels and in SEC experiments (Supplementary Fig. 9c, d).

In order to confirm that trimeric NELL2 can simultaneously bind three Robo3 molecules, we purified a complex of full-length mNELL2 and the mRobo3 FN1 domain via SEC (Supplementary Fig. 10a). The observed elution volume of this complex is indeed consistent with 3:3 stoichiometry, and quantitative western blots show equimolar binding of NELL2 and Robo3, making a 3:1 stoichiometry very unlikely (Supplementary Fig. 10b). We therefore hypothesized that the axon guidance activity of Robo3.1 may depend on NELL2-mediated oligomerization. To test this, we measured commissural axon turning in response to NELL2$^{\Delta CC}$. While there is residual activity, NELL2$^{\Delta CC}$ is severely impaired in its ability to elicit axon repulsion (Fig. 7e, f). We confirmed that the ΔCC construct binds Robo3 FN1–3 with similar affinity as wild-type NELL2, indicating that the CC region does not contribute to NELL2-mediated repulsion via direct interactions with Robo3 (Supplementary Fig. 10c–e). Together, these results support the theory that ligand-mediated receptor oligomerization strongly potentiates NELL2–Robo3 signaling for axon repulsion. Finally, we checked if binding of NELL2 causes large conformational changes in Robo3. Pair distance distribution of mRobo3 ECD with and without mNELL2 EGF1–6 bound show little shape difference, providing evidence against conformational change as a signaling mechanism for NELL2 activation of Robo3 (Supplementary Fig. 10f).

## Discussion

Neuronal connectivity within circuits is shaped by axon guidance cues and their receptors. The Slit–Robo ligand–receptor pair mediates axon repulsion and fulfills a central function in regulating axon guidance at the nervous system midline. Here, we expand on this principal neuronal communication axis with a molecular and functional characterization of the divergent Robo family member, mammalian Robo3, which does not bind Slit proteins but interacts with its own repulsive ligands of the NELL family. Robo3 is essential for commissural axon crossing of the spinal cord midline, and emergence of its unique properties has been proposed as a key evolutionary step in sculpting the mammalian brain[26]. We identify the NELL1/2–Robo3 complex interface and validate its importance for NELL-mediated axon repulsion. We also elucidate the molecular basis of NELL binding preference for Robo3 vs. other Robos and identify a low-affinity NELL2–Robo1 interaction site that is occluded by Robo1 ectodomain architecture. Lastly, we demonstrate that the Robo3.1–not Robo3.2–splice isoform mediates NELL2-induced axon repulsion, and we provide evidence that NELL2-dependent Robo3.1 multimerization strongly contributes to signaling output by the ligand–receptor complex. Our results reveal structural requirements for axon guidance by the NELL–Robo3 complex and shed light on multiple evolutionary specializations of mammalian Robo family members.

We observed in our crystal structures that the FN1 domain of Robo3 is engaged by the NELL1 and NELL2 EGF2 and EGF3 domains using a set of amino acids conserved across vertebrate Robo and NELL sequences. Since we have identified alanine mutations on both EGF domains that can nearly abolish Robo3 binding, it is clear that both domains are required for the ligand–receptor interaction, which agrees with our domain mapping data. Interactions between FN and EGF domains, to our knowledge, are not commonly reported, despite both domains being highly represented in cell surface and secreted proteomes in bilateria. One prominent example of EGF domains interacting with FN domains is Netrin binding to its receptors DCC and Neogenin[42,43], which have domain arrangements similar to

Robos (4 IG followed by 6 FN vs. 5 IG and 3 FN). However, the topology of binding in Netrin–DCC and Robo–NELL complexes are completely unrelated, and the two receptor–ligand pairs appear to have arisen independently, despite functioning on the same set of commissural neurons in the developing spinal cord.

We also demonstrate a direct relationship between affinities of engineered NELL2 variants to Robo3 and the degree of repulsive effect of NELL2 on growing commissural axons in vitro. The correlation between a thermodynamic property of the interaction measured with isolated proteins and the downstream cellular phenotype proves the physiological relevance of the NELL2–Robo3 crystallographic interface and shows the exquisite control NELL2 has on the growth cone as a guidance cue.

Similar to NELL2, we show that NELL1 can elicit commissural axon repulsion in vitro, which can be explained by several observations we report here. First, NELL1 binds Robo3 with an affinity comparable with NELL2. Second, our structural and biochemical analyses of the NELL1–Robo3 EGF1–3–FN1 complex demonstrate highly similar modes of receptor engagement by the EGF2 and EGF3 domains of the two NELL family members. Third, NELL1 and NELL2 share a common oligomeric state—they are both trimeric. Despite these similarities, the guidance activity of NELL1 is at least one order of magnitude lower than NELL2's. This observation suggested the existence of differences in NELL1–Robo3 and NELL2–Robo3 complex architectures that strongly influence downstream signaling activity, possibly through the actions of the LamG and VWC domains of NELLs. However, our axon repulsion assays using chimeras of NELL1 and NELL2 identify the EGF1–3 domains as the major determinant of repulsive activity for these two proteins. This supports the idea that NELL EGF1–3 domains modulate repulsive activity independent of receptor affinity, possibly through conformational control of full-length NELLs that affects Robo signaling. NELL2 is the only NELL family member expressed in the developing spinal cord[27], calling into question whether NELL1 has the opportunity to interact with Robo3 in vivo. Instead, the known functions of NELL1, including those outside the nervous system, might be mediated by signaling through other Robo receptors (see below) via architecturally divergent complexes and signaling strategies.

NELLs were first identified as Robo3 ligands. Here, we find that NELL2 can interact with moderate affinity ($K_D = 14 \mu M$) with the Robo1 FN1 domain. This interaction can be explained by similarities in the NELL-binding surfaces across vertebrate Robo1, Robo2, and Robo3 sequences. However, the NELL2–Robo1 interaction is very weak ($K_D = 350 \mu M$) when studied in the context of the complete Robo1 ECD. This implies that the Robo1 ECD adopts a conformational state in which NELL2 access to the FN1 domain is blocked. No such inhibition was observed for the NELL2–Robo3 interactions. This can be explained by both of our observations that (1) Robo3 adopts an extended, open architecture, where the maximal pairwise distance ($D_{max}$, 31 nm) corresponds to an extended eight-IG/FN domain molecule, and (2) Robo3 ECD is predominantly a monomer, unlike the dimeric/oligomeric Robo1, at least in its unliganded state. As a result, NELL2 binding to an "open-state", monomeric Robo3 is unhindered by any steric clashes, which is not the case in a more compact and dimeric or tetrameric Robo1 structure reported by Aleksandrova et al[16].

Two studies that were published as our manuscript was in preparation, in combination with our data, provide strong clues to conformational states of vertebrate Robos and interactions with NELLs. First, Yamamoto et al.[40] showed that NELL1 can interact with the FN1 domain of Robo2, but not the full Robo2 ECD. This cryptic interaction is similar to the one we describe for NELL2–Robo1 binding, and both are mediated by the same domains. In addition, we are able to provide an accurate picture

of this binding interface through homology with the NELL2–Robo3 structure we present, confirmed via mutagenesis. Second, Barak et al.[17] show that Robo2 adopts a closed-state monomer, where domains IG4, IG5, FN1, and FN2 (domains 4–7) create a compact hairpin-like structure. The NELL-binding surface of FN1 we report in this manuscript is masked in the closed-state Robo2 structure (Fig. 7g), explaining why NELL binding to full-length Robo1/2 is impaired, while the extended Robo3 engages NELLs with higher affinities. Intriguingly, a Robo1 ECD with its FN1 domain swapped against that of Robo3 proved to be strictly monomeric and a high-affinity binder of NELL2. This highlights the importance of the FN1 domain not only in providing high- and low-affinity NELL-binding sites in Robo3 and Robo1, respectively, but also in dictating the oligomeric state of Robos and enabling an open conformation for NELL engagement.

In our work, we measured a 25-fold difference in NELL2 affinity between full-ectodomain Robo1 and the Robo1 FN1 domain. Since NELL2 can only bind to an open-state Robo1, this indicates that the closed-state Robo1 is ~25-fold more stable than its open-state, providing the first estimate of an equilibrium constant for a Robo1 conformational transition (Supplementary Fig. 6c). It is intriguing to speculate that NELLs can act as regulators of Robo1/2 conformational states, e.g., in postcrossing commissural axons, which express high levels of Robo1/2 and navigate in close proximity to the NELL2 expression domain in the spinal cord ventral horn. Since the closed, auto-inhibited state and open, NELL-bound states of Robo1/2 are incompatible, very high concentrations of NELLs could force Robo1 and Robo2 into an open conformation that can either allow formation of active Robo1/2 or facilitate Robo1/2 binding to Slit and signal transduction. Furthermore, a *trans* tetramer proposed by Aleksandrova et al.[16] and a *trans* dimer proposed by Barak et al.[17], as inhibited cell–cell adhesive states for Robo1/2, would also be broken by NELL binding to Robos. The reverse is also possible—Slit-activated Robo1 and Robo2 may become NELL-responsive. Future structural, biophysical, and functional studies are required to differentiate between plausible models and reconcile existing data.

As discussed above, mammalian NELL1 and NELL2 interact with all three neuronal Robos, although with different affinities. The most parsimonious explanation for these observations is that the ancestral molecules in the chordate lineage that gave rise to the mammalian Robo and NELL paralogs interacted with each other. Since the ancestral Robo is also expected to bind Slit, it is conceivable that this receptor or some of the extant Robo family members might engage both Slits and NELLs through high-affinity interaction sites, which needs to be tested. It is also of interest to determine whether invertebrate NELLs can interact with Robos. While we did not observe strong conservation at the Robo interaction surface for invertebrate NELL sequences, we cannot rule out Robo–NELL interactions in extant invertebrates or for an ancestral bilaterian Robo–NELL pair.

Robo3 has been reported to have two splice isoforms, Robo3.1 and Robo3.2, which have identical ECDs and can both bind NELLs; however, spinal commissural axons only respond to NELL2 before crossing the midline, when they express Robo3.1, but they are unresponsive to NELL2 after crossing, at which point they express Robo3.2[27,30]. Through isoform-specific rescue experiments of *Robo3*[−/−] neurons, we show that Robo3.1 is necessary and sufficient for dorsal spinal cord axon repulsion from NELL2. Hence, Robo3.1 is the sole functional NELL receptor in commissural neurons, and distinct signaling capabilities of Robo3.1 and Robo3.2 underlie the differential sensitivity of pre- and postcrossing commissural axons to NELL2. Since Robo3.1 and Robo3.2 only differ in the extreme C-termini

of their ICDs, this suggests that critical mediators of NELL2 signaling are recruited to the unique C terminus of Robo3.1. Robo1 and Robo2 transduce Slit signaling via binding of adapter proteins to the cytoplasmic sequences that are conserved across Robos[3]. It remains to be determined whether NELL2–Robo3 signaling employs at least some of the same cytoplasmic motifs and downstream mediators as Slit–Robo1/2 and whether NELL–Robo1/2 complexes produce functional output, given the absence of a sequence similar to the Robo3.1 C terminus.

A common strategy for signal transduction across the membrane used by single-pass transmembrane proteins is oligomerization upon cue/ligand binding, or the formation of large, lattice-like oligomers, as recently proposed for the Netrin receptor DCC or the protocadherins[42,43,49,50]. As mentioned above, we and others[32,48] have demonstrated that NELLs are constitutively trimeric. Using four different methods—AUC, SEC, MALS, and SAXS—we also demonstrate that their high-affinity unliganded receptor, the ECD of Robo3, is primarily a monomer, while we confirm a predominantly dimeric state for the Robo1 ECD. We also show that the axon guidance activity of monomeric NELL2 lacking its CC domain is strongly reduced. Our results indicate that, unlike Slit-binding Robos, Robo3 does not multimerize in the absence of ligand, and that NELL2-mediated formation of Robo3 trimers or higher-order oligomers is a main driver of Robo3 activation and repulsive signaling (see Fig. 7g for a structural model).

## Methods

**Animals**. All experimental procedures had institutional approval through Brown University's Institutional Animal Care and Use Committee and followed the guidelines provided by the National Institutes of Health. Mice carrying the *Robo3* null allele have been described before and were genotyped by PCR amplifying part of the *Robo3* genomic locus (primer sequences: TACCAGCTACTTCCAGAGAG, CCAACATCGAGTGGTACAAG, and GATCTCTCGTGGGATCATTG)[22]. All mice were maintained on a CD-1 background. For timed pregnancies, the day of vaginal plug was defined as E0.5. Tissue sections or neuronal cultures were prepared from embryos of either sex.

**Spinal cord electroporation and primary neuron culture**. The dorsal fifth of E11.5 mouse spinal cords were microdissected and used for preparation of neuronal cultures; in some instances, dissection was preceded by introduction of expression constructs via spinal cord electroporation. For electroporation, 100 ng/ml DNA in injection buffer (30 mM HEPES pH 7.5, 300 mM KCl, 1 mM MgCl$_2$, 0.1% Fast Green FCF (Sigma)) was injected into the central canal of the neural tube, and a BTX ECM 830 electroporator was used to electroporate DNA bilaterally into spinal cord neurons (Five 30-V pulses of 50 ms duration each for each half of the spinal cord). All rescue constructs are listed in Supplementary Table 5. Dissected spinal cord fragments were washed in cold Ca$^{2+}$/Mg$^{2+}$-free Hanks' Balanced Salt Solution (HBSS). Tissue was digested with 0.05% trypsin in phosphate-buffered saline (PBS; Gibco) at 37° for 7 min, and DNase I was added for an additional 1 min along with 0.15% MgSO$_4$. Tissue pieces were washed with warm Ca$^{2+}$/Mg$^2$-free HBSS, and a small fire-polished Pasteur pipette was used to triturate and dissociate the tissue into single cells. Cells were plated on nitric acid-washed and baked 18-mm coverslips coated with 100 μg/ml Poly-D-Lysine (Sigma) and 2 μg/ml N-Cadherin (R&D Systems). Cells were cultured in Neurobasal media supplemented with 10% heat-inactivated FBS and 1x penicillin/streptomycin/glutamine (Gibco). Neurons were used for experiments 16–24 h after plating. One hour prior to use in axon turning assays, media were switched to Neurobasal media (Gibco) supplemented with 2% B-27 (Gibco) and 1x penicillin/streptomycin/glutamine.

**AP-fusion protein binding experiments**. COS-7 cells (ATCC) were cultured in DMEM media (Gibco) supplemented with 5% fetal bovine serum (Gibco) and 1x penicillin/streptomycin/glutamine (Gibco). COS-7 cells were transfected using Viofectin (Viogene)[27]. The mouse Robo3.1, human Robo3$^{ΔIG1-5}$, and human NELL2-AP and NELL2$^{EGF1–6}$-AP expression constructs have been described before[27]. Information about all other plasmids for protein expression are listed in Supplementary Table 5. Untagged and AP-tagged proteins for AP-fusion protein binding assays and repulsion experiments were expressed in situ following transfection with Viofectin (Viogene). Cells expressing AP-fusion proteins were kept in Opti-MEM (Gibco), and media were collected after 48 h of culture[27]. AP-binding assays were carried out 44–48 h following COS cell transfection. To assess binding

of AP-fusion proteins to receptor-expressing COS cells, cultures were rinsed twice in AP-binding buffer (HBSS (Gibco), 20 mM HEPES pH 7.0, 0.2% BSA, 5 mM CaCl$_2$, 1 mM MgCl$_2$, 2 μg/ml Heparin (Sigma)) and incubated with AP-fusion proteins diluted in AP-binding buffer for 90 min at 4 °C[27]. Cultures were washed three times with AP-binding buffer, and fixed in 4% paraformaldehyde (PFA) in PBS for 15 min at room temperature. Cells were then washed three times with an aqueous solution of 20 mM HEPES pH 7.4, 150 mM NaCl. Endogenous AP activity was deactivated by incubating cultures at 65 °C for 3 h. Cultures were then washed with AP buffer (100 mM Tris pH 9.5, 100 mM NaCl, 50 mM MgCl$_2$), and AP activity was visualized by incubating cultures overnight with NBT/BCIP (Roche; 1:50) in AP buffer. The visualization reaction was stopped by washing cells three times in 1 mM EDTA, 0.1% Triton X-100 in PBS, and cultures were mounted under 80% glycerol in PBS.

**Dunn chamber axon turning assay.** The Dunn chamber axon turning assay was performed essentially as described before[41]. In short, Dunn chambers were removed from a 70% ethanol bath, dried, and washed once with Neurobasal media (Gibco) then twice with conditioned media from primary neuronal cultures. Conditioned media were added to the inner and outer wells of the chamber, and a coverslip containing primary commissural neurons was inverted onto the chamber, leaving a small opening to add or remove liquid from the outer well. Conditioned media were removed from the outer well and replaced with conditioned media containing recombinant NELL1 or NELL2 at specific concentrations. The chamber was then sealed using 1:1 (w/w) paraffin/vaseline. Chambers were kept at 37 °C in a stage-top incubator, and DIC images of neurons on the bridge region of the Dunn chamber (~40 fields per chamber) were acquired every 2 min for 2 h on a Nikon Ti-E microscope.

**Quantification of axon turning.** Quantitative analysis of axon turning in Dunn chambers was performed in a similar fashion to Yam et al.[41]. Briefly, only neurons that extended without encountering debris or other neurons were considered for analysis. Neuron identity was confirmed by post-hoc immunostaining of the imaged coverslip for the commissural neuron marker TAG-1 to differentiate contra- vs ipsilaterally projecting neurons, or for RFP to visualize electroporated neurons. The initial axon segment at $t = 0$ h was translated to the coordinates (0,0) and rotated such that the gradient increased along the $y$-axis (angle of rotation was calculated from microscope stage coordinates relative to the Dunn chamber's center). To determine the magnitude and directionality of an axon's turn, the angle between the original trajectory ($t = 0$ h) of the axon's distal 10 μm and the final trajectory ($t = 2$ h) of the axon's distal 10 μm was calculated. This value, the turning angle, was averaged over 25–100 neurons in each Dunn chamber replicate, and 3–5 biological replicates (tissue from each individual embryo counting as a single replicate) were tested for each experimental condition. Positive angle indicates a turn towards the gradient, and a negative angle indicates a turn away from the gradient. Turning angles were averaged across all neurons for each replicate, and the means across multiple replicates were analyzed for statistical significance using an unpaired two-tailed $t$-test ($n$ and $p$ are indicated in figures and figure legends).

**Immunohistochemistry.** Unless indicated otherwise, all incubations were performed at room temperature. Following imaging in Dunn chambers, coverslips holding the cultured neurons were carefully removed from Dunn chambers and fixed in PBS containing 4% PFA overnight at 4 °C, washed three times for 10 min in PBS, blocked in 2.5% fetal bovine serum and 0.1% Triton X-100 in PBS for 1 h, and incubated with primary antibodies in blocking solution at 4 °C overnight. After three 5 min washes in 0.1% Triton X-100 in PBS, cultures were incubated with secondary antibodies in blocking solution for 2 h. Cultures were then washed three times 10 min in 0.1% Triton X-100 in PBS and mounted on glass slides using Fluoromount G. For cell-surface staining of neurons under nonpermeabilized conditions, primary antibody was added directly in the growth media to live neurons and incubated at 4 °C for 2 h. Cells were then washed three times with PBS, fixed in 4% PFA/PBS for 15 min, washed three times for 10 min in PBS, then incubated with secondary antibodies in blocking solution for 2 h. Cultures were washed three times 10 min in 0.1% Triton X-100 in PBS and mounted on glass slides using Fluoromount G. Tissue section staining was performed as previously described[25]. Primary antibodies used for IHC were goat polyclonal antibodies against Robo3 (R&D Systems, 1:200) and TAG-1 (R&D Systems, 1:200), rabbit polyclonal antibodies against NELL2 (gift from Nakamoto Lab, 1:100)[35], RFP (Rockland, 1:500), and TuJ1 (Biolegend, 1:2000), and a rabbit monoclonal antibody against Robo3.1 (1:200)[30]. Secondary antibodies (Invitrogen; 1:200) were Alexa488-conjugated donkey anti-goat, and Alexa594-conjugated donkey anti-rabbit. Hoechst 33342 (Molecular Probes, 1:1000) was added with the secondary antibodies. All images were acquired on a Nikon Ti-E microscope.

**Protein expression for structural biology.** Mouse and human NELL1, NELL2, Robo1, and Robo3 proteins were expressed using baculoviruses. The mature domains or domain truncations of these genes were cloned into pAcGP67A (BD Biosciences) with C-terminal hexahistidine tags and co-transfected into Sf9 cells (*Spodoptera frugiperda*) with linearized baculovirus BestBac 2.0 (Expression Systems) using the TransIT-Insect transfection reagent (Mirus) according to

manufacturer's instructions. Sf9 cells were cultured in SF900 SFM III (Fisher, 12658-019) with 2 mM L-glutamine (HyClone SH30034.02), 20 μg/ml gentamicin sulfate and 10% fetal bovine serum. For infection of High Five cells with baculoviruses, 2–5 ml of virus-containing conditioned media from Sf9 cultures were added per 1 l of High Five culture.

The produced viruses were used to infect cultures of High Five cells (*Trichoplusia ni*, BTI-Tn-5B1-4) grown in suspension (120 r.p.m. on shakers) at 27–28 °C in Insect-XPRESS media (Lonza, BE12-730Q) with 10 μg/ml gentamicin sulfate (Lonza, 17-518 L). Proteins were expressed for 66 h at 27–28 °C post infection. For producing biotinylated proteins, a C-terminal Avi-tag (GLNDIFEAQKIEWHE) followed by a hexahistidine tag was added to expression constructs. Avi-tagged proteins were biotinylated with the *Escherichia coli* biotin ligase BirA.

Primer sequences used for cloning are provided in Supplementary Table 4.

**Protein purification.** Secreted proteins were purified from media using Ni-NTA Agarose resin (Qiagen), followed by SEC with either Superdex 200 10/300 or Superose 6 10/300 columns (GE Healthcare) in HEPES-buffered saline (HBS: 10 mM HEPES pH 7.2, 150 mM NaCl). Protein complexes were isolated by SEC.

**Protein crystallization.** The purified hNELL2 EGF1–6/hRobo3 FN1–3 complex was concentrated to 15 mg/ml in HBS, and crystallized using the sitting-drop vapor diffusion method in two similar conditions: 17% PEG 3350, 0.4 M NaSCN and 0.1 M HEPES, pH 7.5, 12% PEG 6000. The two crystal forms were washed, dissolved in water, and run on SDS-polyacrylamide gels. Silver staining of the gels revealed that the first crystal form was likely NELL2 EGF1–6 complexed to Robo3 FN1 following proteolysis in the linker joining FN1 and FN2 domains (Supplementary Fig. 2a), and the second crystal form likely contained the remaining FN2 and FN3 domains of Robo3.

The purified hNELL1 EGF1–3/hRobo3 FN1 complex was concentrated to 15 mg/ml in HBS, and crystallized using the sitting-drop vapor diffusion method in 20% PEG 3350, 0.2 M Mg(NO$_3$)$_2$. Crystal screening and optimization was performed using a Mosquito crystallization robot (TTP Labtech).

**X-ray crystallography.** Crystals grown in the first condition (hNELL2 EGF1–6/ hRobo3 FN1 complex) were cryoprotected in 17% PEG 3350, 0.4 M NaSCN, 30% glycerol before being vitrified in liquid nitrogen. The crystals diffracted to ~2.7 Å resolution at Argonne National Laboratory's Advanced Photon Source synchrotron beamline 23-ID-D. MR with the first Fibronectin type III domain model from human Robo2 (PDB: 1UEM) yielded a weak solution with a Z-score of 13 using *PHASER*[51], which was not sufficient to determine phases accurately enough to model the entire structure. Therefore, we prepared selenomethionine-labeled human NELL2 EGF1–6 (hRobo3 FN1 contains no methionines) according to Cronin et al.[52] in baculoviral/High Five cell cultures. The SeMet-labeled complex degraded similarly in crystal drops and crystallized in the original crystallization condition. X-ray diffraction data were collected at the peak absorption wavelength for selenium (12660.3 eV) to ~2.7-Å resolution at the APS beamline 24-ID-C.

Both native and SeMet datasets were reduced using *XDS*[53], and subjected to an MR-SAD (MR with single-wavelength anomalous diffraction) phasing strategy using *phenix.autosol*[54]. The MR solution for the Robo3 FN1 domain was again weak but was sufficient to identify two highly ordered methionine sulfur positions, those in M456 and M465, in the EGF2 domains of hNELL2, which was in turn sufficient to phase the anomalous diffraction dataset and yield interpretable electron density maps. Following initial model building using maps from the anomalous dataset, a native dataset was used for complete model building in *COOT*[55] and reciprocal-space refinement using *phenix.refine*[56]. Within the model, the C-terminal EGF domains had inferior density, likely as a result of flexibility and lack of crystallographic contacts. As a result, the modeling of the sixth EGF domain is incomplete, and the details of coordination of the third calcium binding site is unclear. Validation of the model was performed using tools provided in *COOT* and in *Molprobity*[57] as part of the *PHENIX* package.

Crystals grown in the second condition (hRobo3 FN2–3) were cryoprotected in 12% PEG 6000, 0.1 M HEPES, pH 7.5, 30% glycerol before being vitrified in liquid nitrogen. The crystals diffracted to 1.8 Å resolution at APS beamline 23-ID-D. MR with PDB: 4HLJ using the MoRDA pipeline[58] yielded a strong solution. Model building and reciprocal-space refinement was performed using *COOT* and *phenix. refine*, respectively. Validation of the model was performed using tools provided in *COOT* and *Molprobity* as part of the *PHENIX* package.

hNELL1 EGF1–3/hRobo3 FN1 crystals were cryoprotected by adding 30% glycerol to the crystallization condition. Diffraction data to 1.8 Å were collected at APS beamline 23-ID-D. The structure was solved with MR using the NELL2/ Robo3 structure as a model in *PHASER*. The structural model was built and refined as described above. The positioning of the two FN domains within the crystallographic lattice appears to have caused less well-defined density for parts of the FN2 domain due to flexibility, and higher overall B-factors for FN2 atoms.

**Affinity measurements by cell staining.** The cell staining protocol from Salzman et al.[59] was followed after adopting it for *Drosophila* S2 cells and with modifications. NELL2 and Robo3 ectodomain constructs with an C-terminal FLAG tag are

expressed in *Drosophila* S2 cells by creating a fusion with the transmembrane helix of rat Neurexin-1, using transient transfections with the TransIT-Insect transfection reagent (Mirus) according to manufacturer's instructions. After culturing for 3 days, cells were collected and washed with PBS twice. Final cell pellet was resuspended in PBSB (1x PBS + 0.1% Bovine Serum Albumin), incubated with an iFluor488-coupled anti-FLAG antibody (GenScript, 1 μg/ml) for 30 min at 4 °C and washed with PBSB twice. Next, protein ligand at various concentrations was added onto cells for 30 min at 4 °C, and cells were washed twice with PBSB. Finally, cells were mixed with Cy5-labeled streptavidin for 30 min at 4 °C and washed twice with PBSB. (Expression plasmid for a streptavidin cysteine mutant, and expression, purification and Cy5 labeling protocols were a gift from Michael Birnbaum.) Cells were then run through an Accuri C6 flow cytometer to detect fluorescence at the 488 and Cy5/695 nm channels. Analysis was performed using the Accuri C6 software, where gating was performed to select anti-FLAG positive cells, and binding was measured in the Cy5/695 nm channel as mean fluorescence intensities (MFI) (Supplementary Fig. 3b). Ligand concentrations used for staining vs. 695 nm MFI values were plotted for fitting to 1:1 Langmuir binding isotherms (Fig. 3d, e).

**The extracellular interactome assay (ECIA)**. Robo and NELL constructs were cloned in to pECIA-prey and pECIA-bait plasmids[60]. The assay was performed as described before[46,60]. When mutants of Robo and NELL constructs were tested for binding, the expression levels of the bait and prey were normalized for comparison purposes by quantitation via western blot and dilution of samples at higher concentrations.

**Quantitation by western blot**. Quantitation of purified protein from SEC and secreted protein in conditioned media for ECIA was achieved using western blots against the hexahistidine epitope present at the C-termini of all constructs (antibody: fluorescent His-Tag Antibody with iFluor 488, Genscript, catalog no. A01800, 1:500 dilution). Quantitation of fluorescent signal was performed using a Biorad Chemidoc system and ImageLab version 6 (Biorad). For mNELL2 binding to mRobo3 FN1 in Supplementary Fig. 10, peak fractions from the SEC elutions were loaded onto an SDS-PAGE gel in triplicate, along with samples of mNELL2 and mRobo3 mixed in a 1:1 molar ratio as standards. A linear regression was calculated between concentration and intensity of the standards.

**Surface plasmon resonance (SPR)**. SPR experiments were performed with streptavidin (SA) chips using a Biacore T200, 3000, or 8 K (GE Healthcare) or NLC (Neutravidin) chips on a ProteOn XPR36 (Biorad). Robo3 FN1–3 samples were biotinylated using BirA biotin ligase taking advantage of a C-terminal Avi-tag. hRobo3 FN1–3 was captured on an NLC chip, and mRobo3 FN1–3 was captured on an SA chip. The experiments were performed at 25 °C in 10 mM HEPES, pH 7.2, 150 mM NaCl, 0.05% Tween-20, and 0.1% BSA; lack of BSA resulted in binding curves with strong nonspecific binding which appeared to incorrectly imply high affinity and slow dissociation. All curve fitting was done using the BIAEvaluation, Insight evaluation (GE Healthcare), or Prism (Graphpad) following a 1:1 Langmuir binding model. Prism was used especially when a nonspecific-binding term had to be introduced in data fitting (see below).

mRobo1 ectodomain was captured on a CM5 chip using random amine coupling at a density of ~1,200 R.U. The remainder of the SPR experiment was performed as above. Assuming a range of 50 to 100% active surface, the equilibrium binding responses of mNELL2 ectodomain yield $K_D$ values of 190 to 400 μM (Fig. 6b).

mRobo1 FN1–3 was captured on a Streptavidin Biacore chip via chemical biotinylation of N-terminal amine with N-hydroxysuccinimidobiotin (NHS-Biotin) at pH 6.5. 315 RU of mRobo1 was captured on a Biacore SA chip. The remainder of the SPR experiment was performed as above. The results of the experiment are in Fig. 6e, f. While a theoretical maximum of ~235 R.U. was expected for mNELL2 EGF1–6 binding, the sensorgram data yielded an $R_{max}$ of 143 R.U., predicting a reasonable ~60% active Robo1 surface.

For the Robo chimera and mNELL2$^{ΔCC}$ experiments, mRobo3 ECD, mRobo1$^{Robo3-FN1}$ ECD, mNELL2, and mNELL2$^{ΔCC}$ were captured on a Biacore Series S SA chip after chemical biotinylation with NHS-Biotin performed as above. 167 RU of mRobo3, 177 RU of mRobo1$^{Robo3-FN1}$, 100 RU of mNELL2, and 100 RU of mNELL2$^{ΔCC}$ were captured on the surface. The SPR experiment was carried out with the same buffer conditions as above, with the addition of a regeneration step between each run with 10 mM HEPES, pH 7.2, 150 mM NaCl, 0.05% Tween-20, 0.1% BSA, and 10 mM EDTA in order to remove any nonspecific binding.

When full-length NELL2 was used as analyte against a full-ectodomain Robo surface (i.e., mobile phase, see Supplementary Fig. 8g–i), we observed binding with a fast and a slow component. Since we have no independent evidence for a two-step binding mechanism or heterogenous ligand/analyte, we analyzed the binding data with a 1:1 Langmuir binding isotherm with a linear nonspecific-binding term ($NS \times$ [Analyte]). This yields dissociation constants (0.5–1 μM; Supplementary Fig. 8i, j) closely matching other values we measured for Robo3–NELL2 binding.

Finally, in some of our SPR experiments with NELL2$^{ΔCC}$, which were performed on a newly installed Biacore 8 K, larger-than-usual injection spikes were observed (Supplementary Fig. 10c, d). As affinity analysis was performed based on equilibrium binding, these spikes do not affect the reported $K_D$ values.

A list of all dissociation constants measured by SPR experiments is provided in Supplementary Table 2.

**Analytical ultracentrifugation (AUC)**. Experiments were performed on a Beckman Coulter analytical ultracentrifuge with an An50-Ti rotor. The data were analyzed using the c(s) methodology in SEDFIT[61,62]. Partial-specific volume, density, and viscosity were calculated using SEDNTERP[63]. For analysis, partial-specific volumes of 0.7299 and 0.7322 cm³/g for mRobo1 and mRobo3 samples, respectively, were used. Density and viscosity of the solution were assumed to be 1.005 g/cm³ and 0.01025 Poise. AUC data illustrations were drawn using GUSSI[64].

A pilot, single-concentration experiment was performed at 200,000 g at 20 °C with 2.5 μM mRobo1 and 1.8 μM mRobo3 (Fig. 7a). For the concentration series shown in Supplementary Fig. 6a, c, 400 μl mRobo1 and mRobo3 samples were prepared and placed in 1.2-cm pathlength centerpieces, except for the 100 μl and 9.8 μM, 11.1 μM and 20 μM mRobo3 samples, which were placed in 0.4-cm pathlength centerpieces. Samples were centrifuged at 140,000 *g* for mRobo1 and at 200,000 *g* for mRobo3, at 20 °C.

SV data for mRobo3 revealed a major species at 4 S with a frictional ratio of 1.6–1.7, indicating an elongated shape, and an expected mass of 88 kDa, in agreement with a monomer (Supplementary Fig. 6a). A ~7-S peak observed in mRobo3 samples did not become more prominent with increasing mRobo3 concentrations, indicating that this peak may represent a contaminant or misfolded or aggregated species (Supplementary Fig. 6a). For mRobo1, numerous peaks are observed in c(s) distributions (Supplementary Fig. 6c), including the major 6.6-S peak representing dimeric mRobo1, but also peaks at 4 S and several between 8 and 30 S. The species appear to be in equilibrium, as the monomer wanes with increasing mRobo1 concentrations, and the 8.1-S peak (possibly a tetramer) becomes the second most prevalent peak with increasing concentrations.

**Multi-angle light scattering (MALS)**. MALS experiments were done on a Wyatt HELEOS II scattering detector coupled to an AKTA FPLC (GE Healthcare) with a Superose Increase 6 10/300 column using HBS as the sample and running buffer. For data collection and analysis, ASTRA software version 5.3.4 (Wyatt Technology Corp) was used. Average molar mass values calculated using the Zimm model were 117 kDa for hRobo3 (expected monomer: 108 kDa), 373 kDa for hNELL2 (expected trimer: 296 kDa), 292 kDa for mNELL1 (expected trimer: 301 kDa) and 294 kDa for mNELL2 (expected trimer: 296 kDa).

**Small-angle x-ray scattering (SAXS)**. hRobo3 ectodomain and full-length hNELL2 samples were injected at a concentration of 10–30 μM in a Superose 6 Increase 10/300 size exclusion column equilibrated with HBS at the Advanced Photon Source beamline 18 (Bio-CAT). For mRobo1, mRobo3, and mRobo1$^{Robo3-FN1}$ ectodomains, samples with ~1 mg total of protein were injected. Data reduction and processing were performed using the BioXTAS RAW package[65]. Guinier plots for both datasets were drawn in RAW and P(r) analysis as performed with GNOM. Molecular weights were calculated using the Vc method of Rambo and Tainer[66], yielding a monomeric 100 kDa for hRobo3 ECD and trimeric 283 kDa for hNELL2. Bead models were calculated using DAMMIF[67]. Models provided by DAMMIF were averaged and filtered by DAMAVER and DAMFILT[68]. Bead models shown in Fig. 7c, d are produced from DAMFILT. A list of data statistics for all SAXS analysis is provided in Supplementary Table 3.

**Reporting summary**. Further information on research design is available in the Nature Research Reporting Summary linked to this article.

## Data availability
Further information and requests for reagents may be directed to, and will be fulfilled by the corresponding authors A.J. and E.Ö. The atomic coordinates and crystallographic structure factor amplitudes of the NELL2–Robo3 complex, NELL1–Robo3 complex and Robo3 FN2–3 structure are deposited at the Protein Data Bank under the accession codes PDB: 6POG [https://doi.org/10.2210/pdb6POG/pdb], PDB: 6POK [https://doi.org/10.2210/pdb6POK/pdb], and PDB: 6POL [https://doi.org/10.2210/pdb6POL/pdb], respectively. The source data underlying Figs. 1d, g; 2e; 3d, e; 4a; e; 5c, d; 6f, h; 7f and Supplementary Figs. 3c; 4a, e, i; 5a, b; 6a, b; 8i, 9c; 10b and e are provided as a Source Data file.

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

## Acknowledgements

We thank Nicole Ladd, Yonghong Zhou, and Agnieszka Olechwier for technical assistance, Shouqiang Cheng for critical discussions, and Tian Li and Eduardo Perozo for access to and help with MALS equipment. We acknowledge Anthony Kossiakoff, Hyunh Lee, Elena Solomaha, and the biophysics core facilities at the University of Chicago and University of Illinois at Chicago for access to SPR equipment, and Chad Brautigam and the UTSW Biophysics Core Facility for AUC experiments. We are also grateful to Frederic Charron and Shirin Makihara for help establishing Dunn chamber axon turning assays, and we thank Masaru Nakamoto for generously sharing the NELL antibody. We thank Andrew McCarthy for sharing their Robo1 ECD model, and Srinivas Chakravarthy and Tobin Sosnick for help with SAXS data collection and analysis. This work was supported by NIH grants T32 GM077995 (Z.J.D.), R01 NS097161 (E.Ö.), R01 MH114102 (E.Ö.), RI-INBRE P20 GM103430 (A.J.), and R01 NS095908 (A.J.), a Klingenstein-Simons Fellowship in the Neurosciences and an Alfred P. Sloan Foundation Research Fellowship to E.Ö., and by funding from the Whitehall Foundation to A.J., and by Brown University. This research used resources of the Advanced Photon Source (APS), a US Department of Energy (DOE) Office of Science User Facility operated for the DOE Office of Science by Argonne National Laboratory under Contract No. DE-AC02-06CH11357. This work is partly conducted at a Northeastern Collaborative Access Team beamline, which is funded by the National Institute of General Medical Sciences (NIGMS) from the National Institutes of Health (NIH) (P30 GM124165). The Pilatus 6-M detector on 24-ID-C beamline is funded by an NIH-ORIP HEI grant (S10 RR029205). Data were collected at GM/CA@APS, which is funded by the National Cancer Institute (ACB-12002) and the NIGMS (AGM-12006). The Eiger 16 M detector was funded by an NIH-ORIP HEI Grant (1S10OD012289-01A1). Date were also collected at BioCAT beamline 18-ID, which is supported by grant 9 P41 GM103622 from NIGMS of the NIH. Use of the Pilatus 3 1 M detector was provided by grant 1S10OD018090-01 from NIGMS.

## Author contributions

Conceptualization, A.J., E.Ö.; investigation, J.S.P., Z.J.D., J.W., N.A., Y.P.; formal analysis, J.S.P., Z.J.D., J.W., A.J., E.Ö.; methodology, J.S.P., Z.J.D., J.W., A.J., E.Ö.; supervision, A.J., E.Ö.; visualization, J.S.P., Z.J.D., J.W., A.J., E.Ö.; writing of original draft, A.J., E.Ö.; manuscript review and editing, J.S.P., Z.J.D., J.W., A.J., E.Ö.

## Competing interests

The authors declare no competing interests.
