## [Peer Review File · Nature Communications]

Reviewers' comments:

Reviewer #1 (Remarks to the Author):

Commissural axon guidance is regulated by numerous axon guidance molecules and receptors. NELL2, expressed in the ventral horn, repels commissural axons from the ventral horn via binding to its receptor Robo3. This manuscript by Wang et al. combines detailed structural analysis of NELL2 complexed with its receptor Robo3, with elegant functional analysis in mouse commissural neurons to identify the mechanism for Robo3 receptor activation by NELL2 in axon guidance. They show that NELL2-Robo3 binding is mediated by the EGF2 and EGF3 domains in NELL2 and the FN1 domain of Robo3. Furthermore, they propose that presentation of the FN1 domain and the overall Robo3 extended ectodomain conformation promotes NELL2 ligand accessibility and therefore NELL2 binding. The authors identify residues in NELL2 required for Robo3 binding and show that these residues are also required for NELL2 repulsive activity in commissural neurons. Interestingly, they find that NELL2 repulsive activity is enhanced by ligand trimerization and propose that Robo3 trimerization is important for Robo3 signaling and repulsive activity.

The experiments are well performed and of a high quality, and the manuscript is well written. The manuscript is timely and the findings are new and interesting. However, addition of a few experiments to support their model that "ligand-mediated receptor oligomerization strongly potentiates NELL2-Robo3 signaling for axon repulsion" and further investigation of how the presentation of the Robo3 FN1 domain in the context of Robo1/2/3 ectodomain conformation affects NELL2 ligand binding would strengthen their manuscript for publication in Nature Communications.

Major comments

1) The authors speculate that weaker interaction of NELL2-Robo1 compared to NELL2-Robo3 might be due to different conformations of the Robo receptors. The authors suggest that one reason for the low affinity of NELL2 for Robo1 is that the FN1 interaction site is occluded by the Robo1 ectodomain conformation. To demonstrate this, the authors should do complementary experiments to that in Figure 6, and express Robo1 with its FN1 domain replaced with that of Robo3 and test the ability of this mutant Robo1 (with Robo3 FN1 inserted) to dimerize (demonstrated by analytical ultracentrifugation or size exclusion chromatography, for example) and measure its binding affinity to NELL2. If indeed this mutant Robo1 still dimerizes but is unable to bind to NELL2, it supports their model that the Robo ectodomain conformation dictates how FN1 is presented to NELL2 and will more clearly "elucidate the molecular basis of NELL binding preference for Robo3 vs other Robos" as mentioned in the Discussion.

2a) Whilst the authors show that NELL2 repulsive activity is enhanced by ligand trimerization (Figure 7), they do not show how this affects Robo3 binding and receptor oligomerization. The authors show that deletion of the CC region in NELL2 abolishes NELL2 trimerization and ability to induce a repulsive response in commissural neurons. Although the authors have demonstrated that the EGF2 and EGF3 domains of NELL2 are necessary and sufficient for binding, it does not exclude the possibility that mutations in the CC domain affect binding. The authors should demonstrate whether deletion of the CC region affects the binding affinity of NELL2 to Robo3.

2b) Ideally (but not mandatory), mutation of the 4 cysteine residues, instead of the entire CC region, would be a more specific way to examine the effects of disulfide-dependent trimerization on Robo3 binding and repulsive activity. This will also elucidate whether there are other elements in the CC region that contribute to NELL2 trimerization.

3) The experiments in Figure 7 lead the authors to propose that NELL2 binding to Robo3 leads to Robo3 oligomerization, and that this is important for Robo3 receptor signaling. It is crucial to provide additional evidence to support this, and the Robo3 trimerization/oligomerization in response to NELL2 binding needs to be demonstrated experimentally. Moreover, is there any

conformation change in Robo3 upon NELL2 binding?

Minor comments

4) Second to last paragraph of the Introduction: "While NELLs binds Robo3, they do not strongly interact with full-length Robo1/2." Correct "binds" to "bind".

5) Second to last paragraph of the Results: "as detected using non-reducing, denaturing polyacrylamide gels (Figure 7E). The figure reference is incorrect and should refer to the appropriate supplemental figure.

6) Figure 3(A), legend: hRobo3 should be "green" and hNELL2 should be "cyan".

7) Figure 3(E), legend: States that "hRobo3 FN1" is used for the binding experiments, but the x-axis of the graph states "hRobo3 FN1-3". Please correct.

8) Figure 4(A): x-axis label should read "mNELL2 EGF1-6" to be consistent with the text and figure legend.

9) Figure S4(F), legend: Extra "sites" in the following sentence: "This site is sites conserved among vertebrates and invertebrate NELL" should be deleted. Also, there is a reference to Fig. S3D, which does not exist.

10) This reviewer found that the NELL1 experiments presented in Figure 5 slightly distracted from the main focus on NELL2/Robo3 and suggest that the authors consider placing the data in supplementary data.

11) Can the authors comment on why they think that despite the similarities between the FN1 domains of Robo1 and Robo3, and the slightly weaker affinity of the isolated Robo1 FN1 domain for NELL2, that the Robo3 mutant expressing the Robo1 FN1 domain has no repulsive activity in response to NELL2?

Reviewer #2 (Remarks to the Author):

Remarks to the Author:

The manuscript entitled "NELL2-Robo3 complex structure reveals a novel mechanism of receptor activation for axon guidance" by Wang et al. reports the structure of a minimal NELL2-Robo3 complex and its functional implications on the Robo3 signaling. The authors describe the interaction in detail and carry out a large number of cell based staining and biophysical techniques (SPR, AUC and SAXS) to support the importance of this interaction, and role of the disulfide mediated oligomerization of NELL2. The experiments are described in detail and the results rigorously analyzed so I have confidence in their interpretation, which is consistent across the many techniques used. These results are supported by elegant cell based turning assays on dissected spinal cord neurons to show the functional relevance of a specific NELL2-Robo3.1 interaction. The observation that NELL2 (and 1) can both bind Robo3 using chordate sequence conserved residues is interesting because mammalian Robo3s do not bind Slit2. In addition the higher affinity reported here for Robo3 FN1 vs Robo1/2 FN1 and the known closed conformation of Robo1/2 all support the suggestion that the divergence and evolution of Robo3 from Robo1/2 maybe a key factor in mammalian brain development. The results are new and bring an important contribution to the neuronal development field in general and Robo signaling area in particular. In my opinion the key finding that trimeric NELL2 can induce Robo3 oligomerization for signaling is very worthy of consideration for publication in Nature Communication with some minor revision.

Minor criticisms

1. It would be useful to add the C-terminal deletion of the Robo3.2 alternative splicing isoform to the Robo schematic shown in Fig 1A.
2. The X-ray crystallography experiments are well described and most of the validation metrics are within the expected range, apart from the RSRZ outliers reported for Robo3 FN2-3. These are worryingly high for this resolution (1.8Å) and there is nothing in the text justifying this such as loop disorder. Could the authors please comment?
3. While I agree that NELL may indeed be able to act as a conformation regulator of Robo1/2 at high concentrations I'm not convinced this is functionally relevant, particularly in this context. My main argument is that Pignata et al. (<https://doi.org/10.1101/540690>) recently reported that Robo1 and 2 only appear on the neuronal cell surface during/after midline crossing. Perhaps the authors should edit this section of the discussion accordingly.
4. In Figure 1A schematic the NELL EGF domains are in green and Robo FN domains in magenta. However, in Figure 3A, B and C (as well as Fig.5A) the NELL2 EGF domains are blue and Robo3 FN domain is green. In Fig. S4B the NELL1 EGF domains are magenta and hRobo3 FN yellow. The authors should consider using a consistent color scheme (when possible) for structure figures.
5. I believe a better mechanistic schematic than that shown in Figure 7H would be beneficial to better summarize the signaling mechanism proposed. I think the results are conclusive enough for this. The authors could move panel A, B, C and D to the supplementary information for additional space.

Reviewer #3 (Remarks to the Author):

Wang et al. have carried out a detailed structure-function study of Robo3-NELL signaling. Their major findings are:

- (1) NELL1 and NELL2 bind identically to Robo3, with NELL domains EGF2 and EGF3 wrapping around Robo3 domain FN1. This novel binding mode was observed in two crystal structures and validated by extensive mutagenesis, binding assays, and a functional axon turning assay.
- (2) NELL2 trimerization via its coiled coil domain greatly increases repellent activity.
- (3) NELL1 binds Robo3 with similar affinity as NELL2, but has greatly reduced repellent activity.
- (4) NELL2 also interacts with Robo1, but more weakly than with Robo3 and in a manner that is dependent on Robo1 conformation.

Collectively, these findings represent a significant advance in our understanding of Robo signaling in axon guidance. Slit binding to Robo has been characterized crystallographically (ref. 10) and we are beginning to understand how Slit binding may affect Robo conformation and induce signaling (refs 14 and 33). Robo3 does not bind Slit, however, and Wang et al. show for the first time how Robo3 binds the alternative repulsive cue, NELL2. Their findings thus add an important piece of the puzzle.

The manuscript is clearly written and illustrated with excellent figures. The findings are based on sound experiments and the crystal structures are of high quality. I have only a few comments:

1. The lower biological activity of NELL1 compared with NELL2 is puzzling. Both NELLs are trimers

and bind Robo3 with similar affinity, yet NELL1 is a much more potent repellent. The authors speculate that this may be due to differences in the LamG or VWC domains, but they have made no effort to test this hypothesis. It should be straightforward to make the required domain deletions and NELL1/2 chimeras and test them in the axon turning assay. The data showing NELL1 trimerization should be added to the supplement.

2. It would be helpful to summarize all molecular masses in a table (i.e. calculated, SEC-MALS, SAXS). This important information currently is buried in the Methods section.

3. The most recent results on Robo-Slit signaling should be added to the Introduction. I appreciate that ref. 33 may have been published while this manuscript was in preparation, but it is unsatisfactory that highly relevant results from another lab are only referred to in the Discussion. Regarding Slit-mediated dimerization of Robo (top of page 3), ref. 13 did not show that full-length Slit is a dimer, or that dimerization is important for activity, or that it leads to recruitment of scaffolding proteins. Please rewrite. Regarding alternative mechanisms, ref. 33 should be cited in addition to ref. 14. In the last paragraph of the Introduction, please say **how** the findings are "consistent with two recently published manuscripts".

4. Panels A and C of Figure S6 should have the same x-axis to facilitate comparison. There is an error in the legend of Figure S6: (C,D) and (E,F) are swapped.

5. Panel H of Figure 7 could be improved by drawing a NELL trimer.

We thank the reviewers for their thoughtful comments and suggestions regarding our manuscript. We also thank you for your guidance and interest in considering a revised version of the paper. Below, we provide a detailed response to all comments. Editors'/reviewers' comments are shown in black, responses in blue.

--

Reviewers' comments:

Reviewer #1 (Remarks to the Author):

Commissural axon guidance is regulated by numerous axon guidance molecules and receptors. NELL2, expressed in the ventral horn, repels commissural axons from the ventral horn via binding to its receptor Robo3. This manuscript by Wang et al. combines detailed structural analysis of NELL2 complexed with its receptor Robo3, with elegant functional analysis in mouse commissural neurons to identify the mechanism for Robo3 receptor activation by NELL2 in axon guidance. They show that NELL2-Robo3 binding is mediated by the EGF2 and EGF3 domains in NELL2 and the FN1 domain of Robo3. Furthermore, they propose that presentation of the FN1 domain and the overall Robo3 extended ectodomain conformation promotes NELL2 ligand accessibility and therefore NELL2 binding. The authors identify residues in NELL2 required for Robo3 binding and show that these residues are also required for NELL2 repulsive activity in commissural neurons. Interestingly, they find that NELL2 repulsive activity is enhanced by ligand trimerization and propose that Robo3 trimerization is important for Robo3 signaling and repulsive activity.

The experiments are well performed and of a high quality, and the manuscript is well written. The manuscript is timely and the findings are new and interesting. However, addition of a few experiments to support their model that "ligand-mediated receptor oligomerization strongly potentiates NELL2-Robo3 signaling for axon repulsion" and further investigation of how the presentation of the Robo3 FN1 domain in the context of Robo1/2/3 ectodomain conformation affects NELL2 ligand binding would strengthen their manuscript for publication in Nature Communications.

Major comments

1) The authors speculate that weaker interaction of NELL2-Robo1 compared to NELL2-Robo3 might be due to different conformations of the Robo receptors. The authors suggest that one reason for the low affinity of NELL2 for Robo1 is that the FN1 interaction site is occluded by the Robo1 ectodomain conformation. To demonstrate this, the authors should do complementary experiments to that in Figure 6, and express Robo1 with its FN1 domain replaced with that of Robo3 and test the ability of this mutant Robo1 (with Robo3 FN1 inserted) to dimerize (demonstrated by analytical ultracentrifugation or size exclusion chromatography, for example) and measure its binding affinity to NELL2. If indeed this mutant Robo1 still dimerizes but is unable to bind to NELL2, it supports their model that the Robo ectodomain conformation dictates

how FN1 is presented to NELL2 and will more clearly “elucidate the molecular basis of NELL binding preference for Robo3 vs other Robos” as mentioned in the Discussion.

We have originally had reservations about this experiment, since we thought that a chimera containing Robo1-IG4, IG5 and FN2, and Robo3-FN1 would not be able to create the four-domain, closed hairpin structure (seen in Robo2, Barak et al. 2019) that excludes NELL2 binding.

We have now followed the reviewer’s suggestion and performed this experiment, which supports our original model. Replacement of the FN1 domain in Robo1 with Robo3-FN1 caused a **loss of dimerization** and forced an **open state** to allow strong binding to NELL2 (new Suppl. Fig. 8). The Robo3 FN1 domain is likely not able to create closed, hairpin-like structures with Robo1 IG4, IG5 and FN2 domains. This fully open conformation and monomeric state allows for full access to NELL2 binding.

This result points towards a connection between FN1 domains and dimerization. The existing literature is undecided on if and how the Robo dimers form, and available structural data is in conflict with each other. We intend to follow up in a later manuscript to determine how the FN1 domain and the closed hairpin state may lead to oligomerization in Robo1.

2a) Whilst the authors show that NELL2 repulsive activity is enhanced by ligand trimerization (Figure 7), they do not show how this affects Robo3 binding and receptor oligomerization. The authors show that deletion of the CC region in NELL2 abolishes NELL2 trimerization and ability to induce a repulsive response in commissural neurons. Although the authors have demonstrated that the EGF2 and EGF3 domains of NELL2 are necessary and sufficient for binding, it does not exclude the possibility that mutations in the CC domain affect binding. The authors should demonstrate whether deletion of the CC region affects the binding affinity of NELL2 to Robo3.

2b) Ideally (but not mandatory), mutation of the 4 cysteine residues, instead of the entire CC region, would be a more specific way to examine the effects of disulfide-dependent trimerization on Robo3 binding and repulsive activity. This will also elucidate whether there are other elements in the CC region that contribute to NELL2 trimerization.

We again took upon the reviewer’s suggestion and performed this experiment. If the NELL2 coiled coil region is in direct interaction with Robo3, we expected to see a large loss of binding using the NELL2^{ΔCC} construct against Robo3. We could only observe a 2-fold reduction in binding, which is within expectations due to statistical effects of having three binding sites on the NELL2 trimer (NELL2 is a trimer but NELL2^{ΔCC} is a monomer) (new Suppl. Fig. 10c-e)

We chose not to make a new set of expression viruses for the cysteine mutants. The reason is, even without the disulfides, the trimeric coiled coil is expected to form, since the sequence features of a coiled-coil are strongly present in NELL2 without the cysteines. Removing the cysteines would get rid of disulfides (NELL2^{C→S} mutant would

show up as a monomer on a non-reducing gel), but this protein would likely remain a stable trimer in solution.

3) The experiments in Figure 7 lead the authors to propose that NELL2 binding to Robo3 leads to Robo3 oligomerization, and that this is important for Robo3 receptor signaling. It is crucial to provide additional evidence to support this, and the Robo3 trimerization/oligomerization in response to NELL2 binding needs to be demonstrated experimentally. Moreover, is there any conformation change in Robo3 upon NELL2 binding?

Our efforts to study full-length Robo-NELL complexes in solution via biophysics have been stalled due to the insolubility of this complex. However, we were able to look at fragments of Robo3 binding to full-length NELL2 without solubility issues. We mixed NELL2 with excess Robo3 FN1, and the soluble complex was run over a size exclusion column. Quantitative western blots show that the complex is the result of equivalent molar binding, and SEC elution volumes are compatible with a 3:3 to 6:6 complex, ruling out 3:1 or 1:1 stoichiometries. (new Suppl. Fig. 10a)

Due to the insolubility issue, we have not been able to investigate conformational changes upon complex formation using full-length molecules. However, SAXS experiments with mRobo3 ECD showed no large conformational changes when bound to a truncated mNELL2 (EGF1-6). (new Suppl. Fig. 10f)

Minor comments

We thank the reviewer for his/her attention to detail. The minor comments 4 to 9 have been addressed in the revised manuscript.

4) Second to last paragraph of the Introduction: "While NELLs binds Robo3, they do not strongly interact with full-length Robo1/2." Correct "binds" to "bind".

5) Second to last paragraph of the Results: "as detected using non-reducing, denaturing polyacrylamide gels (Figure 7E). The figure reference is incorrect and should refer to the appropriate supplemental figure.

6) Figure 3(A), legend: hRobo3 should be "green" and hNELL2 should be "cyan".

7) Figure 3(E), legend: States that "hRobo3 FN1" is used for the binding experiments, but the x-axis of the graph states "hRobo3 FN1-3". Please correct.

8) Figure 4(A): x-axis label should read "mNELL2 EGF1-6" to be consistent with the text and figure legend.

9) Figure S4(F), legend: Extra "sites" in the following sentence: "This site is sites conserved among vertebrates and invertebrate NELL" should be deleted. Also, there is a reference to Fig. S3D, which does not exist.

10) This reviewer found that the NELL1 experiments presented in Figure 5 slightly distracted from the main focus on NELL2/Robo3 and suggest that the authors consider placing the data in supplementary data.

We strongly considered this suggestion. However, since we had to include additional data and supplements for NELL1/2 chimeras, as requested by reviewer #3, we decided against moving NELL1 entirely to the supplement.

11) Can the authors comment on why they think that despite the similarities between the FN1 domains of Robo1 and Robo3, and the slightly weaker affinity of the isolated Robo1 FN1 domain for NELL2, that the Robo3 mutant expressing the Robo1 FN1 domain has no repulsive activity in response to NELL2?

We have two explanations for this observation. First, Robo1 FN1 affinity for NELL2 is ~25-fold weaker than that of Robo3 FN1. This alone can cause a very significant loss of repulsive activity. Second, Robo1 exists in an oligomeric and occluded state, where Robo1 FN1 domain is not available for NELL binding in the context of the full-length Robo1. Conversely, Robo3 is mostly in a monomeric and open state that allows NELL2 to bind freely.

These two factors result in a very weak NELL2 affinity for Robo1 ECD. The overall K_D is approximately 350 μM , which is too weak for NELL2 to illicit any repulsive activity through Robo1.

Reviewer #2 (Remarks to the Author):

The manuscript entitled “NELL2-Robo3 complex structure reveals a novel mechanism of receptor activation for axon guidance” by Wang et al. reports the structure of a minimal NELL2-Robo3 complex and its functional implications on the Robo3 signaling. The authors describe the interaction in detail and carry out a large number of cell based staining and biophysical techniques (SPR, AUC and SAXS) to support the importance of this interaction, and role of the disulfide mediated oligomerization of NELL2. The experiments are described in detail and the results rigorously analyzed so I have confidence in their interpretation, which is consistent across the many techniques used. These results are supported by elegant cell based turning assays on dissected spinal cord neurons to show the functional relevance of a specific NELL2-Robo3.1 interaction. The observation that NELL2 (and 1) can both bind Robo3 using chordate sequence conserved residues is interesting because mammalian Robo3s do not bind Slit2. In addition the higher affinity reported here for Robo3 FN1 vs Robo1/2 FN1 and the known closed conformation of Robo1/2 all support the suggestion that the divergence and evolution of Robo3 from Robo1/2 maybe a key factor in mammalian brain development. The results are new and bring an important contribution to the neuronal development field in general and Robo signaling area in particular. In my

opinion the key finding that trimeric NELL2 can induce Robo3 oligomerization for signaling is very worthy of consideration for publication in Nature Communication with some minor revision.

Minor criticisms

1. It would be useful to add the C-terminal deletion of the Robo3.2 alternative splicing isoform to the Robo schematic shown in Fig 1A.

This helpful suggestion has been implemented in the revised figure.

2. The X-ray crystallography experiments are well described and most of the validation metrics are within the expected range, apart from the RSRZ outliers reported for Robo3 FN2-3. These are worryingly high for this resolution (1.8Å) and there is nothing in the text justifying this such as loop disorder. Could the authors please comment?

The reviewer brought up a point we did not explain in the manuscript. The RSRZ values for amino acids in the Robo3 FN2-3 are not evenly distributed. Out of the 45 outliers, 44 are on the FN2 domain. This is due to the fact that electron-density for the Robo3 FN2-3 structure varies in quality. The third FN3 domain is well-restrained by crystal contacts, and has low B-factors (average = 32 Å²), but the second FN3 domain is more mobile, and electron density resembles that from diffraction at ~2.5 Å (average B = 67 Å²). This is reflected in the RSRZ statistic:

Figure. A,B. Robo3 FN2-3 structures colored by RSRZ (A) or B-factors (B). **C.** $2mFo-DFc$ electron density map for Robo3 FN2-3. FN3 is restrained by crystallographic contacts and has strong density. FN2 is less restrained, and density is less well defined (see red arrows), resulting in RSRZ outliers in the FN2 domain (A). **D.** $2mFo-DFc$ electron density map for Robo3 FN2 drawn at 0.8σ .

However, it should still be stated that the FN2 domain is built based on solid evidence. When electron density is contoured at lower sigma levels, the features built are well justified. In further support, no amino acid has a map Correlation Coefficient of less than 0.7, and the geometry of the model is highly regular.

The methods section of the revised manuscript now states these issue for the Robo3 FN2-3 and the Robo3-NELL2 structure, which also has significant disorder for the EGF6 domain.

3. While I agree that NELL may indeed be able to act as a conformation regulator of Robo1/2 at high concentrations I'm not convinced this is functionally relevant, particularly in this context. My main argument is that Pignata et al. (<https://doi.org/10.1101/540690>) recently reported that Robo1 and 2 only appear on the neuronal cell surface during/after midline crossing. Perhaps the authors should edit this section of the discussion accordingly.

We fully agree that the functional relevance needs to be established for this theory. However, NELLs are strongly present near neurons that express Robo1 and Robo2. For example, post-crossing commissural axons, which express high levels of Robo1/2, navigate in close proximity to the NELL2 expression domain in the spinal cord ventral horn.

4. In Figure 1A schematic the NELL EGF domains are in green and Robo FN domains in magenta. However, in Figure 3A, B and C (as well as Fig.5A) the NELL2 EGF domains are blue and Robo3 FN domain is green. In Fig. S4B the NELL1 EGF domains are magenta and hRobo3 FN yellow. The authors should consider using a consistent color scheme (when possible) for structure figures.

We attempted to use a consistent scheme for coloring structures across the manuscript. NELL2 and NELL1 are depicted cyan and magenta, respectively. Consecutive NELL2 EGF domains, which needed to be distinguished in Fig. 2, use light cyan to blue in a continuum of colors. We need to differentiate Robo3 FN1 bound to NELL2 and to NELL1, since we overlay these two in Fig. 5A: therefore Robo3 FN1 is green (bound to NELL2) and yellow (bound to NELL1).

However, we failed to extend this to the schematic in Fig. 1A. We thank the reviewer for pointing this out. This is now remedied in the revised Figure 1, and further extended in the new model schematic added to the paper (new Fig. 7g).

5. I believe a better mechanistic schematic than that shown in Figure 7H would be beneficial to better summarize the signaling mechanism proposed. I think the results are conclusive enough for this. The authors could move panel A, B, C and D to the supplementary information for additional space.

We recognize the need for a better mechanistic model, which has now been added to the manuscript (new Fig. 7g).

Reviewer #3 (Remarks to the Author):

Wang et al. have carried out a detailed structure-function study of Robo3-NELL signaling. Their major findings are:

(1) NELL1 and NELL2 bind identically to Robo3, with NELL domains EGF2 and EGF3 wrapping around Robo3 domain FN1. This novel binding mode was observed in two crystal structures and validated by extensive mutagenesis, binding assays, and a functional axon turning assay.

(2) NELL2 trimerization via its coiled coil domain greatly increases repellent activity.

(3) NELL1 binds Robo3 with similar affinity as NELL2, but has greatly reduced repellent activity.

(4) NELL2 also interacts with Robo1, but more weakly than with Robo3 and in a manner that is dependent on Robo1 conformation.

Collectively, these findings represent a significant advance in our understanding of Robo signaling in axon guidance. Slit binding to Robo has been characterized crystallographically (ref. 10) and we are beginning to understand how Slit binding may affect Robo conformation and induce signaling (refs 14 and 33). Robo3 does not bind Slit, however, and Wang et al. show for the first time how Robo3 binds the alternative repulsive cue, NELL2. Their findings thus add an important piece of the puzzle.

The manuscript is clearly written and illustrated with excellent figures. The findings are based on sound experiments and the crystal structures are of high quality. I have only a few comments:

1. The lower biological activity of NELL1 compared with NELL2 is puzzling. Both NELLs are trimers and bind Robo3 with similar affinity, yet NELL1 is a much more potent repellent. The authors speculate that this may be due to differences in the LamG or VWC domains, but they have made no effort to test this hypothesis. It should be

straightforward to make the required domain deletions and NELL1/2 chimeras and test them in the axon turning assay. The data showing NELL1 trimerization should be added to the supplement.

We followed the reviewer's comments and created NELL1-NELL2 chimeras where the EGF1-3 domains were swapped between NELL1 and NELL2. The chimeras can still bind Robo3, but only the NELL1^{NELL2 EGF1-3} has repulsive activity. This implies that the LamG and VWC domains do not play a major role in repulsive activity of NELLs. (new Fig. 5d and Suppl. Fig. 5)

Literature contains NELL1 trimerization data in the form of non-reducing SDS-PAGE gels. We supplemented these by in-solution trimerization data for natively folded NELL1 using MALS (new Suppl. Fig. 9b).

2. It would be helpful to summarize all molecular masses in a table (i.e. calculated, SEC-MALS, SAXS). This important information currently is buried in the Methods section.

We have added these tables to the manuscript (new Suppl. Tables 2 and 3).

3. The most recent results on Robo-Slit signaling should be added to the Introduction. I appreciate that ref. 33 may have been published while this manuscript was in preparation, but it is unsatisfactory that highly relevant results from another lab are only referred to in the Discussion. Regarding Slit-mediated dimerization of Robo (top of page 3), ref. 13 did not show that full-length Slit is a dimer, or that dimerization is important for activity, or that it leads to recruitment of scaffolding proteins. Please rewrite. Regarding alternative mechanisms, ref. 33 should be cited in addition to ref. 14. In the last paragraph of the Introduction, please say *how* the findings are "consistent with two recently published manuscripts".

Per the reviewer's request, we now mention the work of Yamamoto et al. and Barak et al. in the Introduction. The literature for how Slit signals through Robo is contentious, and the exact biochemical nature of active and inactive Slit-Robo complexes is unsettled. We extended the introduction by citing more of the literature and emphasizing the presence of numerous models.

We recognize that ref. 12 (Howitt 2004) first demonstrated the Slit dimer formation (which was later dismissed by the same group in Hohenester 2008). We had cited ref. 13 (Seiradake 2009) not for the discovery of the dimer, but for the high-resolution structure of the dimerization domain. In the revised manuscript, this section is now rewritten.

4. Panels A and C of Figure S6 should have the same x-axis to facilitate comparison. There is an error in the legend of Figure S6: (C,D) and (E,F) are swapped.

We have the AUC data for Robo1 and Robo3 displayed on the same plot in in Fig. 7a for an easy comparison. The supplements mentioned are intended to show in detail the concentration series used in these experiments.

The swapped figure legends (now in supplemental Fig. 7) have been fixed. We thank the reviewer for pointing this out.

5. Panel H of Figure 7 could be improved by drawing a NELL trimer.

Unfortunately, we do not have a structure of the NELL trimer, and the coiled coil is separated from the EGF domains by two VWC domains. We do not know much about the structural conformations of the NELL trimers, and we are not sure how to draw it explicitly into structural models. In our models, we have indicated trimerization via “x3” signs.

REVIEWERS' COMMENTS:

Reviewer #1 (Remarks to the Author):

In this revised manuscript, the authors have addressed all my comments, with new experimental data where possible. Overall, the new data is well integrated into the revised manuscript and demonstrates the key role of the Robo FN1 domains in receptor oligomerization and binding to NELL2, and shows that NELL2 as a trimer can bind simultaneously to three Robo3 FN1 fragments. These results clarify the molecular basis of the NELL2 and Robo3 interaction, strengthen their conclusions and improve their manuscript. I congratulate the authors on a beautiful study and recommend this manuscript for publication in Nature Communications.

Reviewer #2 (Remarks to the Author):

The authors have addressed all my concerns in the revised manuscript and/or rebuttal response so I'm happy to recommend publication after a small clarification (see below). The new experimental results presented further validate the original conclusions and improve the overall quality of the manuscript, as well as providing new insights on the stability of the Robo receptor family.

One worrying aspect of the new results are the SPR figures in Suppl. Fig. 8 (g and h), and particularly Suppl. Fig. 10 (c and d). All have breaks in the SPR curves after injection ($t=0$) and elution ($t=60$). Similar effects are not present in the other SPR curves (eg. Suppl. Fig 4 c and d, or g and h). This maybe a format/drawing mistake and should be corrected. If these originate for some other cosmetic reason it should be reported and justified in the SPR methods section.

Reviewer #3 (Remarks to the Author):

The authors have successfully addressed all of my concerns. Their review of the literature now is well balanced and they have carried out the requested experiment using NELL1/2 chimeras. It is interesting that the outcome of this experiment was not the expected one.

They have responded to the other reviewers' comments in a similarly constructive manner.

The revised manuscript is greatly improved and I am happy to recommend publication.

Reviewer #1 (Remarks to the Author):

In this revised manuscript, the authors have addressed all my comments, with new experimental data where possible. Overall, the new data is well integrated into the revised manuscript and demonstrates the key role of the Robo FN1 domains in receptor oligomerization and binding to NELL2, and shows that NELL2 as a trimer can bind simultaneously to three Robo3 FN1 fragments. These results clarify the molecular basis of the NELL2 and Robo3 interaction, strengthen their conclusions and improve their manuscript. I congratulate the authors on a beautiful study and recommend this manuscript for publication in Nature Communications.

Reviewer #2 (Remarks to the Author):

The authors have addressed all my concerns in the revised manuscript and/or rebuttal response so I'm happy to recommend publication after a small clarification (see below). The new experimental results presented further validate the original conclusions and improve the overall quality of the manuscript, as well as providing new insights on the stability of the Robo receptor family.

One worrying aspect of the new results are the SPR figures in Suppl. Fig. 8 (g and h), and particularly Suppl. Fig. 10 (c and d). All have breaks in the SPR curves after injection ($t=0$) and elution ($t=60$). Similar effects are not present in the other SPR curves (eg. Suppl. Fig 4 c and d, or g and h). This maybe a format/drawing mistake and should be corrected. If these originate for some other cosmetic reason it should be reported and justified in the SPR methods section.

Reviewer #3 (Remarks to the Author):

The authors have successfully addressed all of my concerns. Their review of the literature now is well balanced and they have carried out the requested experiment using NELL1/2 chimeras. It is interesting that the outcome of this experiment was not the expected one.

They have responded to the other reviewers' comments in a similarly constructive manner.

The revised manuscript is greatly improved and I am happy to recommend publication.

Response:

We would like to thank all three reviewers for their constructive and positive comments. Similarly, we are pleased with the latest and improved version of our manuscript, which we hope will be a foundational study on the function of NELLs as Robo ligands, and Robo signaling in axon guidance.

Reviewer #2 has brought up an issue that has also concerned us: Some of the recent SPR data added appears to have sensorgrams broken at injection points. This is the result of our use of the latest SPR machine and software released by GE, the Biacore 8K and the accompanying data analysis software. We observed higher-than usual injection spikes with this new instrument; the data analysis/export software chose to blank out these spikes and allowed us to export sensorgrams with brakes at the injection points. While esthetically unpleasing, these issues do not effect the analysis, since we are only analyzing steady-state binding, which is accurately measured by this highly sensitive and state-of-the-art SPR machine. This is now mentioned in the revised manuscript.